# The Structural and Functional Differences between Three Species of Fish Scale Gelatin and Pigskin Gelatin

**DOI:** 10.3390/foods11243960

**Published:** 2022-12-07

**Authors:** Jinmeng He, Jian Zhang, Yingjie Xu, Yigang Ma, Xiaobing Guo

**Affiliations:** School of Food Science and Technology, Shihezi University, North Fourth Road, Shihezi 832003, China

**Keywords:** pigskin gelatin, fish scale gelatin, chemical structure, functional properties

## Abstract

In this paper, gelatin was extracted from the scales of *Coregonus peled*, *Carp* and *Bighead carp* by the acid method, and the structure and functional properties of the obtained scale gelatin and food-grade pigskin gelatin (FG) were compared. The results showed that all gelatins exhibited relatively high protein (86.81–93.61%), and low lipid (0.13–0.39%) and ash (0.37–1.99%) contents. FG had the highest gel strength, probably because of its high proline content (11.96%) and high average molecular weight distribution. Low β-antiparallel was beneficial to the stability of emulsion, which led FG to have the best emulsifying property. The high content of hydrophobic amino acids may be one of the reasons for the superior foaming property of *Bighead carp* scales gelatin (BCG). The gel strength of *Carp* scales gelatin (CG) and BCG, the ESI of *Coregonus peled* scales gelatin (CPG) and the foaming property of BCG indicate that fish gelatin has the potential to be used in food industry as a substitute for pig skin gelatin.

## 1. Introduction

Gelatin, a denatured protein derived from collagen through thermo-hydrolysis, is widely used in food, pharmaceutical, cosmetic and photographic applications because of its emulsifying, gelling, film forming and foaming properties [1]. Currently, due to the increasing application of gelatin in the food and beverage, pharmaceuticals and nutraceuticals industries, the global demand for gelatin is increasing [2]. In 2020, the global gelatin market reached nearly $3.2 billion, with an annual output of about 620 kilotons, mainly produced in Western Europe. According to the forecast of Grand View Research, by 2027, the global total production of gelatin will reach about 870 kilotons, worth 6.7 billion USD, at an average annual growth rate of 9.2%. Gelatin is mainly obtained from by-products of mammals; it can be divided into pigskin gelatin (PSG) (46.00%), bovine hide gelatin (29.40%), bovine bone gelatin (23.10%) and other gelatins (1.50%), according to its source. However, the application of mammalian gelatin is limited by consumer concerns about the spread of bovine spongiform encephalopathy, foot-and-mouth disease, hand-foot-mouth disease and other diseases. Additionally, due to special dietary needs, the application of mammalian gelatin is limited [3,4]. Therefore, the gelatin industry is interested in developing alternative gelatins from other mammals, such as camels, and from non-mammals, such as fish and insects (e.g., sorghum worms and melon worms) [5]. The application of gelatin from camels and insects is limited because of a lack of sources [6]. Compared with mammalian and insect gelatin, fish gelatin has the advantages of abundant sources, no risk of zoonotic diseases and no special dietary requirements [7,8].

The fishery industry is an important economic sector. In 2018, the total catch in the world was about 179 million tons, and this is increasing year by year. In the process of production and processing, a large amount of fish waste is produced. According to statistics, the waste produced by fish processing may be as high as 85.00% of the total catch [9]. This fish waste is usually discarded or processed into products with low commercial value (fish meal, feed and fertilizer) [10]. Therefore, finding a new way to increase the economic benefit of fish waste has become an urgent problem for the fishery industry. Fish scale is one of the waste products resulting from fish processing; it accounts for about 3% of the total weight of fish and is rich in collagen. Therefore, using fish scales to prepare gelatin not only avoids the problems of food safety and some special dietary needs to a certain extent, but also contributes to the high-value utilization of this waste material.

Considering the potential commercial value of fish scale gelatin (FSG), its functional properties have attracted the attention of researchers in recent years. Sha et al. [11] studied the effect of ammonium sulfate precipitation on the gelatin gel properties of bighead carp scales. Sha et al. [12] studied the emulsifying and foaming properties of bighead carp scale gelatin treated by dynamic high-pressure micro-jet technology. Xu et al. [13] studied silver carp scale gelatin for the stabilization of fish oil-loaded emulsions. Gelatin extracted by Tkaczewska et al. [14] from carp skin by three pretreatment methods had the potential for industrial application as an alternative source of mammalian gelatin. Gelatin extracted by Chandra et al. [15] from three kinds of freshwater fish skin had worse textural characteristics than that from pig skin. In this work, we selected FG and FSGs (CPG, CG and BCG) as research subjects, and the FSGs were extracted by the acid method. The structural and functional properties of the different gelatins (FG, CPG, CG, and BCG) were compared. This work will provide useful information to understand the relationship between the structure and function of gelatin.

## 2. Materials and Methods

### 2.1. Materials

FG was purchased from Zhejiang Yinuo Biological Technology Co., Ltd. (Ningbo, China). Fish scales were obtained from fishery companies in different regions of Xinjiang, China and transported to the laboratory by refrigerated trucks. *Coregonus peled* fish scales were selected from Xinjiang Sailimu Lake Fishery Technology Development Co., Ltd. (Bole, China). *Carp* scales and *Bighead carp* scales were selected from Xinjiang Ruixue Aquatic Products Breeding Co., Ltd. (Altay, China). Colza oil was purchased from a local supermarket. All chemicals were analytical grade.

### 2.2. Extraction of Fish Scale Gelatins

The obtained fish scales were washed with distilled water, naturally dried and shattered using a grinder for 1 min. The samples were stored in a desiccator. Fish scale gelatins were extracted according to the method described by Sha et al. [11] with slight modifications. For the decalcification of fish scales, 0.5 mol/L HCl was used in a material-to-liquid ratio of 1:10, and mechanical stirring was carried out at 1000 rpm for 1 h, performed twice. After the decalcification was completed, the fish scales were washed with distilled water and filtered with gauze until the pH value of the supernatant was close to that of distilled water. Deionized water was added in a ratio of 1:3 (*v*/*w*) to fish scales, and the pH of the mixture was adjusted to 5.5 with 1 mol/L NaOH. The scales were incubated at 80 °C for 2 h. The scale residue was removed by centrifugation at 5000 rpm for 20 min, and the supernatant was passed through 400 mesh filter clothes and lyophilized for use. The gelatin extraction rate was calculated according to the following formula:(1)$$E\left(\%\right)=\frac{m}{M}\times 100\%$$
where *E* is the extraction rate of fish scale gelatin (%); *m* is the mass of gelatin extract (g); and *M* is the mass of dried fish scale (g).

### 2.3. The Gelatins Composition

Moisture, lipid, ash and protein contents were calculated by standard methods (AOAC, Singapore, 2000). The nitrogen content in fish scale gelatin was measured by the Kjeldahl method (KjelMaster k-375, BUCHI, St. Gallen, Switzerland), and the protein conversion factor was 5.55.

### 2.4. Determination of Color

The color of gelatins was measured by a colorimeter. The instrument was calibrated using a standard white and black board. We filled the powder test box with gelatin and tightened the lid, and then measured with a corrected colorimeter. We subsequently recorded the values of L* (brightness), a* (red-green value, where + is red and − is green) and b* (yellow-blue value, where + is yellow and − is blue). Each test was performed in triplicate.

### 2.5. Turbidity

The turbidity of gelatin was analyzed according to the method proposed by Zhang et al. [16] with slight modifications. Briefly, 10 mg/mL gelatin solutions were prepared at different pH values (3, 5, 7, 9, and 11). FG-3, FG-5, FG-7, FG-9 and FG-11 represent the FG solution at pH 3, pH 5, pH 7, pH 9 and pH 11, respectively. CPG-3, CPG-5, CPG-7, CPG-9 and CPG-11 represent the CPG solution at pH 3, pH 5, pH 7, pH 9 and pH 11, respectively. CG-3, CG-5, CG-7, CG-9 and CG-11 represent the CG solution at pH 3, pH 5, pH 7, pH 9 and pH 11, respectively. BCG-3, BCG-5, BCG-7, BCG-9 and BCG-11 represent the BCG solution at pH 3, pH 5, pH 7, pH 9 and pH 11, respectively. We transferred the gelatin solution into a 20 mL glass bottle, placed it neatly on a horizontal table with black cloth at the back and took photos with a digital camera under indoor light. We then placed 3 mL gelatin solution into a glass cuvette and measured the absorbance at 600 nm with a UV-vis spectro-photometer. Ultrapure water was used as the blank control.

### 2.6. Solubility

The gelatin solutions (2 mg/mL) were centrifuged at 5000 rpm for 20 min at 4 °C, and the gelatin concentration in the supernatant was determined by biuret method. The solubility was calculated according to the following formula:(2)$$S\left(\%\right)=\frac{c}{C}\times 100\%$$
where *S* is the solubility (%); *c* is the concentration of the supernatant (mg/mL); and *C* is the original concentration (mg/mL).

### 2.7. Chemical Structure

#### 2.7.1. Sodium Dodecyl Sulfate—Polyacrylamide Gel Electrophoresis (SDS-PAGE) Research

SDS-PAGE analysis was performed according to the method described by Díaz-Calderón et al. [17]. At room temperature (25 ± 1 °C), gelatin solutions (2 mg/mL) were mixed with 2× loading buffer at a ratio of 1:1. One group of samples was reduced and the other was non-reduced. The samples were denatured at 100 °C for 5 min and then centrifuged at 10,000× g for 1 min after cooling. Next, 10 μL of the sample was injected into 5% concentrated gel, and the sample was separated with 7.5% separating gel. The analysis was carried out at 80 V for 50 min and then analyzed at 120 V for 3 h. The protein band was stained with Coomassie Brilliant Blue R-250 for 45 min and decolorized with 10% acetic acid for 4–5 h. Finally, the bands were visualized with a gel imaging system.

#### 2.7.2. Amino Acid Composition

The amino acid composition of gelatin was analyzed according to the method described by Sila et al. [18] with slight modifications. The gelatin was dissolved in ultrapure water, and 10 mL HCl (6 mol/L) was added to 1 mL gelatin solution (20 mg/mL), which was then mixed and placed in an ice-water bath. We then added 3~4 drops of phenol solution, blew in nitrogen and further hydrolyzed at 110 °C for 24 h. After hydrolysis, the sample was filtered into a 50 mL volumetric flask, and after a constant volume was reached, 1 mL solution was transferred into a test tube, which was blown to dryness with nitrogen. Next, we added 1 mL ultrapure water into the test tube, blew dry with nitrogen and dissolved 1 mL sodium citrate buffer (0.2 mol/L). The mixture was then filtered through a 0.22 μm filter membrane and injected into an amino acid analyzer (L-8900, Hitachi, Tokyo, Japan).

#### 2.7.3. FTIR Spectrum Analysis

The FITR spectra of gelatin was analyzed according to the method described by Yang et al. [19]. Gelatins and KBr were mixed in a ratio of 1:100 and pressed into thin slices. The FITR spectra were recorded between 400 and 4000 cm^−1^ on a Fourier transform infrared spectrophotometer (Bruker Vertex 70 V, Shanghai, China). The secondary structure ratios were analyzed by SeaSolve PeakFit V4.04 software. Five regions (amide A, amide B, amide I, amide II and amide III) were analyzed by Omnic software 1.70.

#### 2.7.4. Determination of Endogenous Fluorescence Spectroscopy

The endogenous fluorescence spectroscopy of the gelatin solutions was analyzed according to the method described by Yang et al. [19], with slight modification. Gelatins were dissolved in 10 mmol/L phosphate buffer solution with a pH of 7.0, and the test sample solution containing 0.2 mg/mL gelatin was prepared. The fluorescence spectrometer (S8 TIFER, Bruker, Germany) was operated with an excitation wavelength of 295 nm and a slit width of 10 nm. The fluorescence emission spectra were collected in the range of 260–350 nm.

#### 2.7.5. Determination of Surface Hydrophobicity Index (H_0_)

The surface hydrophobicity index of the gelatin solutions was measured according to the method described by Yang et al. [19]. Briefly, the ammonium salt of 1- anilino -8-naphthalenesulfonate (ANS) was used as a fluorescent probe. Gelatin was prepared in solutions with different concentrations (0.02 mg/mL, 0.05 mg/mL, 0.1 mg/mL, 0.2 mg/mL and 0.5 mg/mL, respectively). We then made ANS into 8 mM solution with the same phosphate buffer solution and stored it away from light for later use. Next, 40 μL ANS solution and 4 mL gelatin solution were mixed evenly, and the fluorescence intensity of the mixed solution was detected by fluorescence spectrometry (S8 TIFER, Bruker, Germany). Parameters: excitation wavelength 390 nm, emission wavelength 470 nm, slit width 10 nm and test temperature 25 °C. The difference between the fluorescence intensity of gelatin solution with ANS and that without ANS was the relative fluorescence intensity. The relative fluorescence intensity of gelatin solution was plotted against the intensity of gelatin solution, and the initial slope of the curve was the surface hydrophobicity index of the gelatin to be measured.

### 2.8. Functional Properties

#### 2.8.1. Gelatin Gel Strength and Texture Profile Analysis (TPA)

##### Determination of Gelatin Gel Strength

According to the method of GME (with slight modifications), a gelatin solution with a concentration of 6.67% was prepared to measure the gel strength [15]. The gelatin solution was poured into a 25 mL glass beaker and left to mature at 4 °C for 16–18 h [20]. The gel strength of gelatin gels was measured using a texture analyzer (TA. X T Plus, Stable Micro Systems, Surrey, UK) equipped with a cylindrical aluminum probe (P 0.5 R). The gel strength was recorded as the maximum force (g) when the penetration distance reached 4 mm at a speed of 1 mm/s.

##### Texture Profile Analysis

The 6.67% gelatin solutions were poured into molds and incubated at 4 °C for 16–18 h. The gels were removed and cut it into small cylindrical colloids with a diameter of 1.5 cm and a height of 2 cm. The TPA of gelatin gels were measured using a texture analyzer (TA. X T Plus, Stable Micro Systems, Surrey, UK) equipped with a flat cylindrical probe (P 36 R). The gels were subjected to two cycle compressions until they were compressed to 40% of their original heights at a speed of 1.0 mm/s.

#### 2.8.2. Determination of Emulsifying Properties

##### Emulsifying Activity Index (EAI) and Emulsifying Stability Index (ESI) Analysis

The EAI and ESI of gelatin solutions were measured according to the method of Nagarajan et al. [21], with slight modifications. Colza oil was added to the gelatin solution with concentration of 1 mg/mL (*w*/*v*) and pH 9 in a ratio of 1:3 in a 100 mL beaker. The mixture was homogenized at 15,000 rpm with a high-speed shearing machine for 2 min. We then transferred 50 μL of the emulsion to a test tube containing 5 mL sodium dodecyl sulfate (SDS) solution with concentration of 0.1% (*w*/*v*), mixed it well and measured the absorbance of the diluted emulsion with an ultraviolet spectrophotometer at 500 nm. EAI and ESI were calculated using following the equations:(3)$$EAI\left(m^{2}/g\right)=\frac{2\times 2.302\times A_{0}\times DF}{c\times \varphi \;\times 10000}$$
(4)$$ESI\left(min\right)=\frac{A_{0}\times 10}{A_{0}-A_{10}}$$
where *A*_0_ and *A*_10_ are the absorbance of the diluted emulsion after homogenization at 0 and 10 min, respectively; *DF* is the dilution factor; *c* is the weight of protein per volume (g/mL); and φ is the volume fraction of oil in the emulsion.

##### Particle Size and ζ-Potential Analysis of Emulsions

According to the method described by Huang et al. [22] with slight modification, the ζ-potential and particle size of the emulsions were determined. The primary emulsion was prepared via the method described in above section, and then the primary emulsion was homogenized for two cycles at 1200 bar using a high-pressure homogenizer. The final emulsion was diluted 100 times to measure its ζ-potential and particle size by particle electrophoresis (Nano-plus, Micromeritics, New York, NY, USA).

##### Creaming Index (CI) Analysis

The emulsions prepared in above section were transferred to a glass bottle, stored at room temperature for 7 days and photographed with a digital camera. In addition, during the emulsion storage process, the height (*H_s_*) of the serum layer (at the bottom of the vials) and the total height (*H_e_*) of the emulsions were measured. Then, the creaming index (*CI*) of the emulsion was calculated using the following equation:(5)$$CI\left(\%\right)=\frac{H_{s}}{H_{e}}\times 100\%$$

#### 2.8.3. Foaming Properties Analysis

The foaming capacity (FC) and foaming stability (FS) of gelatins were measured according to the method described by Salem et al. with some modifications [23]. The sample solution (10 mg/mL) was placed in a 50 mL plastic centrifuge tube with a scale, and the volume of the sample was recorded as *V*_0_. The sample solution was homogenized using a high-speed shearing machine at 15,000 rpm for 1 min, and the sample volume was recorded as *V*. After 10 min, the sample volume was recorded as *V*_1_. *FC* and *FS* were calculated using the following equations:(6)$$FC\left(\%\right)=\frac{V-V_{0}}{V_{0}}\times 100\%$$
(7)$$FS\left(\%\right)=\frac{V_{1}-V_{0}}{V-V_{0}}\times 100\%$$

#### 2.8.4. Water Holding Capacity (WHC) and Fat Binding Capacity (FBC) Analysis

According to the method described Zhang et al. with slight modifications [16], 0.10 g or 0.25 g of gelatin was placed in a centrifuge tube. Ultrapure water (5 mL) or colza oil (2.5 mL) was added, and the tube was vortexed for 10 s every 10 min. After 1 h, the gelatin solution was centrifuged at 450 r/min for 20 min. The upper phase was gently removed, and the remaining gelatin in the centrifuge tube was filtered on 800 mesh filter paper for 30 min. The WHC and FBC were calculated according to the following formula:(8)$$WHC/FBC\left(\%\right)=\frac{M}{m}\times 100\%$$
where *m* is the weight of the dried gelatin (g), and *M* is the weight of the contents in the centrifuge tube after draining (g).

### 2.9. Statistical Analysis

Statistical analyses were carried out using Origin 2022. All of the experiments were random, and samples were measured three times in parallel. The results are presented as the mean values *±* standard deviation. Significant differences between means (*p* < 0.05) were determined through Duncan’s test by one-way ANOVA using SPSS 26.0 (SPSS Inc., Chicago, IL, USA).

## 3. Results and Discussion

### 3.1. Gelatin Compositions

The extraction yield, chemical composition, and solubility of samples are presented in Table 1. The extraction yield of three kinds of FSG was as follows: CG > BCG > CPG. The difference in extraction yield may be linked to many factors, such as age, breed and living environment [18]. The protein content of three kinds of FSG was above 90%. The higher the protein content of gelatin, the lower the other impurities, which is beneficial to the quality of gelatin [24]. In Chinese National Standard GB6783-2013 (Food additive: gelatin) [25], the moisture and ash content of gelatin as a food additive should be lower than 14% and 2%, respectively. There is no requirement for fat. Gelatin with a moisture content between 6% and 8% is considered to be highly hygroscopic, which may complicate the determination of its physical properties [26]. In this study, the moisture content of the three kinds of FSG was less than 6%. The ash content and fat content of the four gelatins were less than 2% and 0.4%, respectively. This shows that these three FSGs reach the standards for these three physical and chemical indexes. Compared with FG, three kinds of FSG have much lower ash content, which indicates that the three kinds of fish gelatin extracted in this paper were of better quality [27]. The high ash content of FG may be due to different extraction methods. There are two main types of gelatin. Type A is derived from collagen with exclusively acid pretreatment, while type B is the result of alkaline pretreatment of collagen. The FG was type B, while the three kinds of FSG were type A. Gelatins prepared by the two different extraction methods show different physical and chemical properties [28].

### 3.2. Physical Properties

#### 3.2.1. Color Value Analysis

The color analysis of four samples is shown in Table 1. The lightness (L*) value of four gelatin samples was in the range of 87.74–90.19, and CG had the highest L* value. The L* value from European eel skin gelatin was 89.28, which was also in this range, and it’s a* and b* were −0.12 and 11.70, respectively [18]. The a* (redness) and b* (yellowness) values of four gelatins follow: FG > CPG ≈ CG ≈ BCG and FG > CPG > CG ≈ BCG, respectively. The a* indicates that FG was red, and the three kinds of FSG were green. The b* indicates that the four kinds of gelatins were yellow, and the FG has the highest b* value. Furthermore, the colors of CG and BCG were not significantly different. The application of gelatin has different color requirements, but the color does not affect the functional properties of gelatin [18].

#### 3.2.2. Solubility Analysis

The solubility of protein can reflect its degree of aggregation. In addition, it can also affect important functional properties of protein [29]. The solubility of gelatin mainly depends on its amino acid composition, protein structure and the lengths of peptides. As shown in Table 1, the solubility of the four samples was in the range of 89.47–94.19%. FG had the lowest solubility (89.47%); CPG had the highest solubility (94.19%); and the three kinds of FSG showed similar values. The better solubility of FSGs may be related to their lower subunit content and more exposed hydrophilic groups, which may promote the balance of hydrophobicity and hydrophilcity, as well as the hydratability of protein to water [19,30]. The low solubility of FG may be associated with low purity.

#### 3.2.3. Turbidity Analysis

As shown in Figure 1A, FG, CG and BCG became turbid at pH 5, pH 9, and pH 9, respectively. However, CPG exhibited no obvious turbidity at any pH range. As can be seen from the Figure 1B, the absorbance of CPG has no significant difference at all on the pH values. This result may be due to the fact that a low content folding structure increases contact with hydrophilic molecules, so that the protein has good solubility [31]. Among the four kinds of gelatin, CPG had the lowest content of β-sheet structure (Figure 2B). SDS-PAGE (Figure 3) showed that CPG had more subunits with lower molecular weight distribution than the other three kinds of gelatin, which may be the reason why it showed better solubility at all pH values. A previous revealed that low molecular weight subunits are beneficial to the solubility of protein [32]. The maximum absorbance values of FG, CG and BCG were at pH 5, pH 9 and pH 9, respectively. This is consistent with the results shown in Figure 1A. Previous studies have shown that protein solutions have maximum turbidity at the isoelectric point; the isoelectric points of type A gelatin and type B gelatin were in the range of 4.8–5.2 and 6.0–9.0 [26,33]. Therefore, FG had an isoelectric point around pH 5, while the isoelectric point of CG and BCG was around pH 9. Among them, CPG exhibited good solubility, even at the isoelectric point.

### 3.3. Chemical Structure

#### 3.3.1. SDS-PAGE

The subunit composition and relative molecular weight distribution of gelatin samples were determined by SDS-PAGE. Many factors affect the molecular weight distribution of gelatin, such as the extraction method, temperature and time. Gelatin is composed of different polypeptide chains, including α chains (90–110 kDa), β chains (180–220 kDa) and γ chains (270–300 kDa). The molecular weight distribution of gelatin has a great influence on its functional properties [5,34]. The electrophoresis results of gelatin are shown in Figure 3. There were significant differences in the molecular weight distribution between FG and the three kinds of FSG, which may be caused by different sources and methods of gelatin extraction. FG had a large molecular weight distribution, and their molecular weights are mostly concentrated in the lane mouth. In contrast, for the three kinds of FSG, the molecular weight of the two α (α1, α2) chains was between 100–150 kDa, and the molecular weight of the β chain was approximately 250 kDa. Moreover, the reduced and non-reduced samples had similar structures, which indicates that there was no disulfide bond between the gelatin molecules. These phenomena are also similar to our previous research [19].

#### 3.3.2. Amino Acid Composition

Amino acid composition is one of the main factors affecting the physical and chemical properties of gelatin. It is mainly characterized by the multiple repetitions of a ‘‘Glycine-X-Y″ sequence, where “X” is mostly Pro, and “Y” is usually Hyp (sometimes Ala) [3,35,36]. As shown in Table 2, CPG had the highest total amino acid content in all samples. Glycine is the most important amino acid of four kinds of gelatin, accounting for about a quarter of the total amino acid content. In addition, the amino acids with higher content are Glu, Ala, Pro, and Arg. But for four kinds of gelatin. The contents of these four amino acids are slightly different. For FG, the contents of four amino acids are as follows: Pro > Glu > Ala > Arg. FG contains the highest content of Pro probably because it is type A gelatin [37]. For CPG, the contents of four amino acids are as follows: Glu > Ala > Pro > Arg. The contents of four amino acids, CG BCG, are the same: Ala > Pro > Glu > Arg. The amino acids with lower content in four kinds of gelatin are Thr, Val, Met, Ile, Leu, Tyr, Phe, and His. These results are consistent with the amino acid composition of gelatin extracted from carp scales and reef cod skin [10,38]. The amino acid contents of four kinds of gelatin are quite different, such as Ser, Met, and His. The Ser and Met contents of CPG are about 2.6 and 8 times of FG, respectively. The His content of CPG is also the highest among the four gelatins. These differences may be caused by different sources of gelatin.

#### 3.3.3. ATR-FTIR

As shown in Table 3 and Figure 2A, there were five amide regions in the four gelatins, namely amide A (3600–3200 cm^−1^), amide B (3100–3000 cm^−1^), amide I (1700–1600 cm^−1^), amide II (1550–1600 cm^−1^) and amide III (1300–1200 cm^−1^, N-H deformation and C-N stretching) [20]; the infrared spectra of the four gelatins had similar peaks. The amide A is related to the stretching vibration of -NH, and when the -NH group is involved in hydrogen bonding, the position will move to a lower wave number [18]. This may be the reason why the band wavelength of FG and BCG amide A is lower than that of CPG and CG. The amide B is related to the stretching of CH_2_- and -NH, and FG and CG showed the lower wavelength compared to CPG and BCG revealed that may be an interaction between the polypeptide chains through -NH_3_ group [10]. The amide I is related to the stretching of C=O, and the secondary structure of gelatin was analyzed by calculating the area of amide I. The amide II is produced by bending vibration of NH- and stretching of CN-. The amide III represents the binding peak between C-N tensile vibration and N-H deformation of amide bond, and there is little difference in the peak position of four kinds of gelatin in this range.

In particular, the secondary structure of gelatin was analyzed by calculating the area of 1700–1600 cm^−1^ (amide I) by software: The content of its secondary structure was calculated by software: 1610–1640 cm^−1^ (β-sheet), 1640–1650 cm^−1^ (random coil), 1650–1658 cm^−1^ (α-helix), 1660–1680 cm^−1^ (β-turn), and 1680–1700 cm^−1^ (β-antiparallel), and the functional groups and secondary structure of gelatin were analyzed by ATR-FTIR. As shown in Figure 2B, the four gelatins exhibited similar α-helix and β-sheet content, and the three kinds of FSG exhibited similar β-antiparallel content. FG had the highest random coil content (10.95%) and CPG had the highest β-turn (29.03%) and β-antiparallel content (12.11%). In addition, the main secondary structures of all gelatins were β-sheet and β-turn, and their content exceeded 66.28%. It is reported that β-folding has an important influence on the interaction with neighboring protein molecules [19].

#### 3.3.4. Fluorescence Spectroscopy

Fluorescence Intensity reflects the tertiary structure of gelatin by measuring the exposure of aromatic amino acids to water. Tryptophan residues are usually located in the core of protein and have high fluorescence intensity. The fluorescence spectrum of gelatin is shown in Figure 4A. The peak positions of four kinds of gelatin are all at 290 nm, but their fluorescence intensities are different, with the following magnitude: BCG > CG > FG > CPG. The willingness to cause this difference may be due to their different tertiary structures. The higher fluorescence intensity of BCG may be due to the fact that more aromatic groups are exposed to water and it is easier to emit fluorescence. The low fluorescence intensity of CPG may be due to the aggregation of some nonpolar amino acid residues covered by protein [39].

#### 3.3.5. Surface Hydrophobicity Index (H_0_)

The exposure degree of hydrophobic amino acids in protein chains can be measured by surface hydrophobicity. The surface hydrophobicity of protein has an important influence on its surface properties. ANS, as a fluorescent probe, can be non-covalently bound to hydrophobic areas on the surface of protein [40]. As shown in Figure 4B, the H_0_ of FG, CPG, CG, and BCG was 11.73, 45.03, 30.07, and 30.44, respectively. CPG had the highest H_0_, which indicates CPG had more exposed hydrophobic regions than the other gelatins. CG and BCG had similar H_0_ value, and FG had the lowest surface hydrophobicity. It was found that cold-water fish usually showed higher hydrophobicity, while warm-water fish contained higher proline content and showed good hydrophilicity [2]. Similar to the results of this paper, *Coregonus peled* is a kind of cold-water fish, and the gelatin extracted from it has the lowest proline content among the four kinds of gelatin. In addition, the aggregation of protein will reduce the surface area of hydrophobic groups exposed to water, resulting in the decrease of surface hydrophobicity [39]. It can be found from Figure 1A,B that the absorbance of CPG is the lowest at pH 7, indicating that the aggregation degree of protein in this solution is low. This may be another reason for its high surface hydrophobicity.

### 3.4. Gel Properties

Gel strength is one of the indexes to evaluate the quality of gelatin. Typically, gelatin from a fish source will have, lower gel strength than from a pig source [15]. Similar results have been obtained in this study, the gel strength and hardness values of the four gelatins (Table 4) were similar and followed the trend: FG > BCG ≈ CG > CPG. This may be related to FG had the highest content of proline, and studies show that amino stabilized the ordered conformation of gel network during gelation. In addition, the reason for this result may be related to the molecular weight distribution. As shown in the Figure 3, the average molecular weight of four kinds of gelatin was: FG > BCG > CG > CPG. Studies show that the high gel strength of gelatin was associated with the high average molecular weight [6,41]. Springiness can be used to characterize the rubber-like elasticity of colloid in mouth and the influence of initial deformation on colloid structure. In this study, the springiness of four kinds of gelatin has similar results. Cohesiveness can be used to measure the integrity of the network structure of the gel after extrusion and can be used for comparative analysis of the influence of the first extrusion and the second extrusion on the gel structure. Among the four kinds of gelatin, CPG has the lowest cohesiveness, which indicates that the network structure of the gel is destroyed most seriously after extrusion, and chewiness can represent the taste of human mouth when chewing. Generally, chewiness is consistent with the change trend of gumminess and hardness of the gel [42], which is consistent with the results of this study.

### 3.5. Emulsifying Properties

Emulsifying is one of the important functional characteristics of gelatin in food application. Emulsifying includes EAI and ESI. Previous studies have shown that the H_0_ of gelatin has a positive effect on the emulsifying of gelatin. Fish gelatin with high H_0_ promotes hydrophobic interactions among proteins at the interface and proteins with oil droplets. This leads to a better hydrophilic-lipophilic balance and promotes the adsorption of protein onto the oil-water interfacial layer and the dispersion of oil droplets [22,43]. Furthermore, the content and impurities of protein also affect the emulsifying ability. The longer the peptide chain, the higher the molecular weight, and the more stable the protein membrane on the interface. The high solubility of protein can quickly disperse to the surface of oil droplets, thus improving the emulsifying efficiency of protein [24,44]. Unfolding of protein can improve emulsion stability by forming films with high surface elasticity and low surface tension [12]. As shown in Figure 5A, the EAI and ESI of four kinds of gelatin as followed: FG (351.90 m^2^/g) > CPG (266.38 m^2^/g) > BCG (240.28 m^2^/g) ≈ CG (225.08 m^2^/g) and FG (71.42 min) > CPG (65.64 min) > CG (14.34 min) ≈ BCG (13.86 min), respectively. 

The particle size of the emulsion can directly reflect the emulsifying properties of protein [45]. The particle size of the four kinds of gelatin follows: BCG (653.30 nm) ≈ CG (627.70 nm) > CPG (552.95 nm) > FG (428.00 nm) (Figure 6A). Huang et al.’ s research shows that the smaller the absolute value of the charge, the less electrostatic repulsion between droplets, thus promoting the flocculation of droplets into large particle size [22]. As shown in Figure 6B, the absolute value of the *ζ*-potential follows: FG (−25.10 mV) > CPG (3.03 mV) > CG (−1.53 mV) ≈ BCG (−1.51 mV). Therefore, FG emulsion has a smaller particle size because it has a higher negative charge and can resist the aggregation of droplets. On the contrary, BCG and CG have the lowest absolute charge, which makes their aggregation resistance the worst and their particle size the largest. At a certain pH value, different gelatin will show different emulsifying stability. On the oil-water interface formed by emulsifying, protein will gather and interact by itself, forming a viscoelastic membrane to prevent the flocculation of droplets. In this study, the emulsions were prepared at pH 9. The *ζ*-potential of the four samples is shown in Figure 6B. The droplets of CPG emulsion were positively charged, indicating that their pI > 9, and the other three gelatin emulsion droplets were negatively charged, indicating that their pI < 9. The FG is a type B gelatin, and its isoelectric point is lower than the three kinds of FSG. The greater the absolute value of potential, the greater the electrostatic repulsion. Therefore, the emulsion prepared from FG had better stability. The difference between FG and FSG may be due to the extraction method, amino acid composition, and protein conformation [12,40]. In addition, the structure of protein determines its functional characteristics. FG has the highest emulsion stability, which may be due to its lower β-antiparallel percentage (Figure 2B) in the secondary structure [16]. For the three kinds of FSG, CPG had the highest EAI, and this might be due to its higher solubility and surface hydrophobicity, which increases the protein concentration at the oil-water interface and the compactness of the interface layer [29,46].

In the process of creaming, the aggregated droplets move upward and gather on the top of the cream, eventually forming a cream layer. As shown in Figure 5B, the CI of the four kinds of gelatin demonstrated the following trend: FG (57.94%) > BCG (47.88%) ≈ CG (45.51%) ≈ CPG (44.55%). The results show the CI depends on the initial size of the emulsion droplets, the interface thickness, and the viscosity of the continuous phase [16]. The three FSG emulsions had clear serum layers after being stored for seven days (Figure 5C), which indicates that the emulsion layer and serum layer of FSG emulsion have more significant drop concentration difference. In other words, the serum layer of FSG emulsion has a lower emulsion quantity and larger drop size, which is consistent with the particle size data [40].

### 3.6. Foaming Properties

Foaming is of great significance to the application of gelatin in foods such as marshmallows. As shown in Figure 7A, the foam capacity of CPG was 115.00%, while those of FG, CG and BCG were 160.00%, 145.00%, and 162.50%, respectively. The foaming stability of FG, CPG, CG and BCG was 40.20%, 26.54%, 27.74%, and 46.21%, respectively. In addition, CG and BCG had similar foaming activity to FG, which indicates that these two kinds of warm water fish scale gelatins may be good alternatives to mammalian gelatin in the application of foaming performance. The difference of foaming property among all gelatins may be influenced by the content of hydrophobic amino acids, the source of protein, the inherent properties and composition of protein, and the conformation of protein at the air/water interface in solution. Jellouli et al. found that the foaming properties of grey triggerfish skin gelatin is slightly higher than that of bovine gelatin, which may be due to the higher content of hydrophobic amino acids in grey triggerfish skin gelatin [18,24]. This study has similar results, the total contents of hydrophobic amino acids (Ala, Val, Ile, Leu, Pro, Met, Phe, Tyr) in four kinds of gelatin are FG (31.28%), CPG (29,93%), CG (31.44%) and BCG (32.03%). The highest content of hydrophobic amino acids in BCG may be one of the reasons for its best foaming properties. The FC and FS of CPG were the lowest, which may be due to the weak protein-protein interaction in the matrix around the foam. In addition, its low viscosity is not conducive to maintaining the thickness of the interface film, resulting in poor foam stability [47].

### 3.7. WHC and FBC

WHC and FBC are important functional characteristics of protein in food system, and they indicate the ability of protein to bind water and fat and maintain them under gravity [16]. As shown in Figure 7B the WHC of FG, CPG, CG and BCG are 1049.59%, 121.17%, 294.18%, and 385.74%, respectively. The FBC of four kinds of gelatin are BCG (641.52%), CG (615.51%), CPG (610.63%) and FG (264.88%), respectively. Furthermore, the WHC and FBC of the four gelatins showed opposite trends. FG had a high WHC, but FBC is weak. There was no significant difference in the FBC of the three FSGs, and they all exhibited high FBC because of their high surface hydrophobicity. This result is similar to that of gelatin from bones of red snapper and grouper [10,41]. WHC was positively correlated with the content of Hyp and Pro. Thus, the high Hyp content and low surface hydrophobicity of FG may be related to its high WHC [3,24,44,48].

## 4. Conclusions

In this paper, the structure and function of four kinds of gelatin were compared. The structure was affected by different sources and extraction technologies. SDS-PAGE showed that all three FSGs had typical α and β chains of gelatin. CPG had the highest total amino acid content and the highest H_0_. The endogenous fluorescence spectrum showed that BCG contained more exposed aromatic amino acids. Among the four kinds of gelatin, FG had the best gel strength, emulsifying properties and WHC; however, its FBC was the smallest. Among the FSGs, CPG had the lowest gel strength and the best emulsifying properties. CG and BCG had similar gel strengths and emulsifying properties. BCG had similar FC as FG and higher FS than FG. CG and CPG had similar foaming properties, albeit lower than those of FG. Therefore, FSG has the potential to replace pig skin gelatin in the food industry. Notably, BCG could be applied as a foaming agent in food. However, the fishy smell and other weak functional properties of fish gelatin limit its application. Therefore, it is necessary to further explore deodorization technology for fish scale gelatin and improve its functional properties.

## Figures and Tables

**Figure 1 foods-11-03960-f001:**
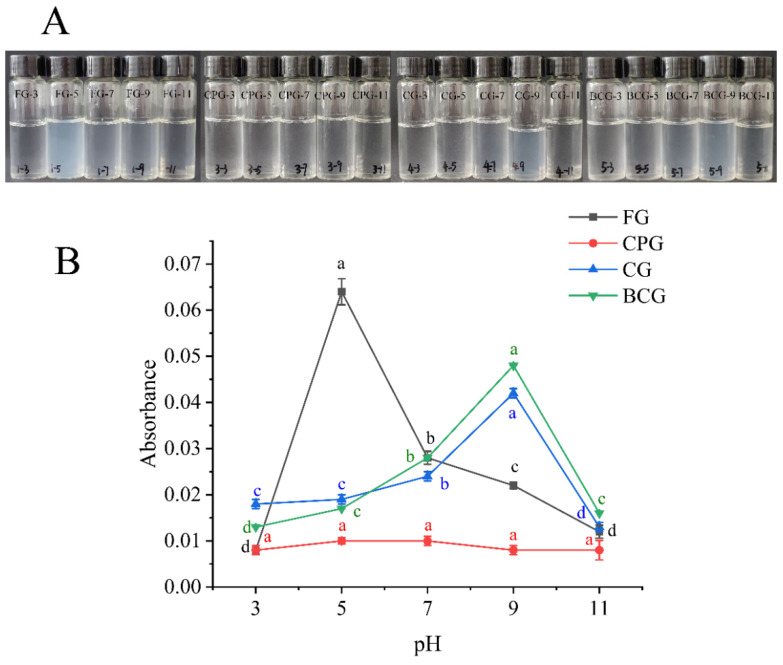
Turbidity analysis of solution; particle size, potential and CI analysis of emulsion. (**A**): Photographs of four gelatins solutions at different pH. (**B**): Turbidity of four gelatins solutions. Different lowercase letters denote significant differences (*p* < 0.05).

**Figure 2 foods-11-03960-f002:**
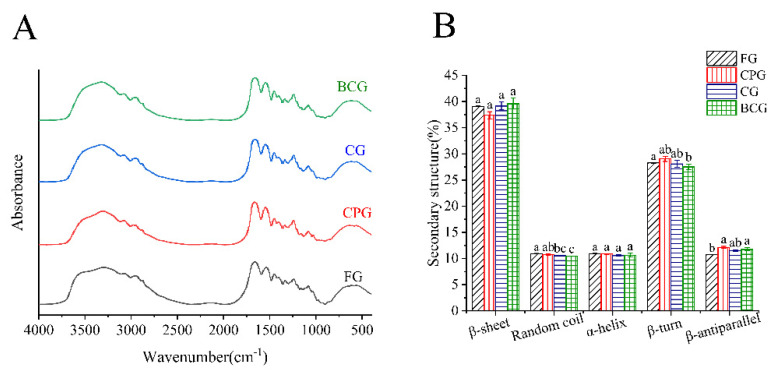
(**A**): ATR−FTIR spectra. (**B**): Secondary structure percentage (%) of four samples. Different lowercase letters denote significant differences (*p* < 0.05).

**Figure 3 foods-11-03960-f003:**
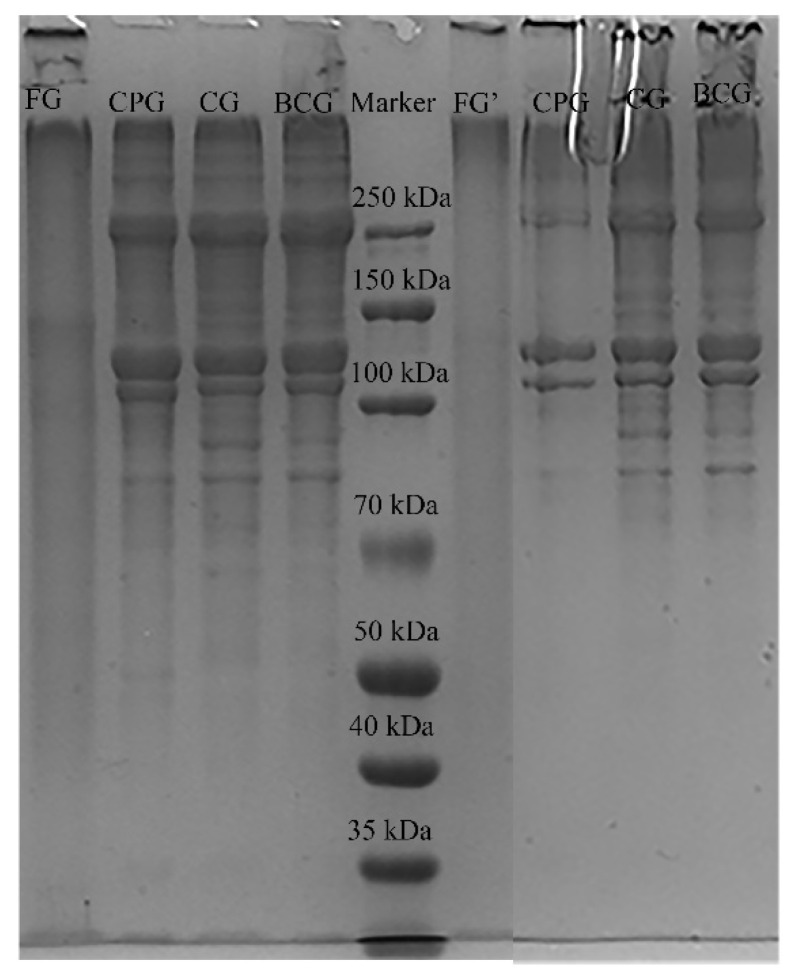
SDS-PAGE pattern of four samples. Maker is protein standard; FG, CPG, CG and BCG are reduced; FG’, CPG’, CG’ and BCG’ are non-reduced.

**Figure 4 foods-11-03960-f004:**
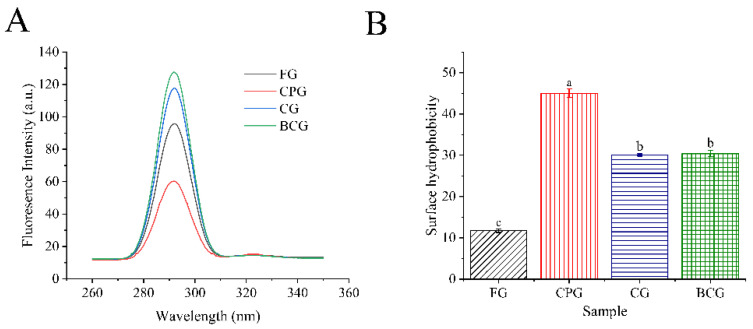
(**A**): Endogenous fluorescence spectrum. (**B**): Surface hydrophobicity index of four samples. Different lowercase letters denote significant differences (*p* < 0.05).

**Figure 5 foods-11-03960-f005:**
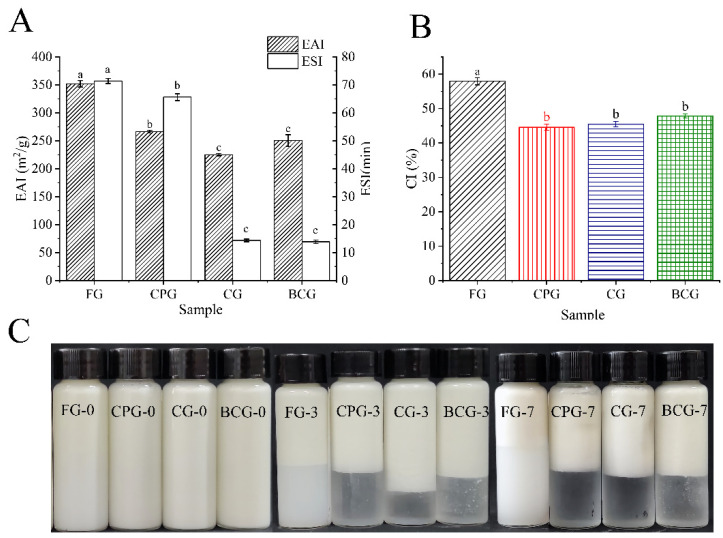
(**A**): EAI and ESI results of four samples. (**B**): Creaming index of four gelatins emulsions. (**C**): Photographs of four gelatins emulsions at different Storage time (FG-0, CPG-0, CG-0, BCG-0: FG emulsion, CPG emulsion, CG emulsion and BCG emulsion on the 0th day of storage respectively. FG-3, CPG-3, CG-3, BCG-3: FG emulsion, CPG emulsion, CG emulsion and BCG emulsion on the 3th day of storage respectively. FG-7, CPG-7, CG-7, BCG-7: FG emulsion, CPG emulsion, CG emulsion and BCG emulsion on the 7th day of storage respectively.). Different lowercase letters denote significant differences (*p* < 0.05).

**Figure 6 foods-11-03960-f006:**
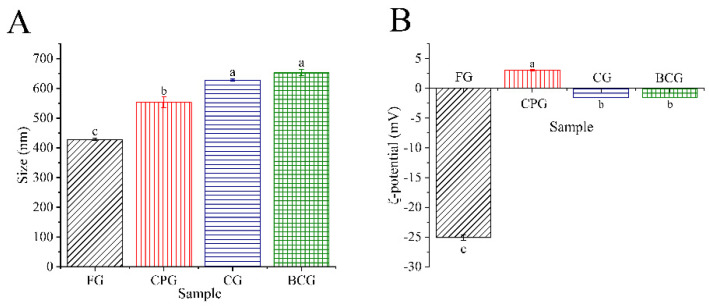
(**A**): Size of four gelatins emulsions. (**B**): ζ−potential of four gelatins emulsions. Different lowercase letters denote significant differences (*p* < 0.05).

**Figure 7 foods-11-03960-f007:**
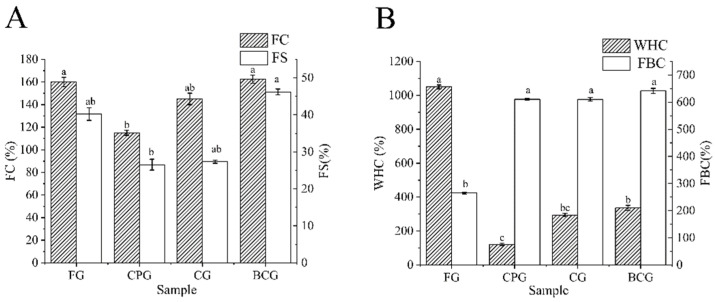
(**A**): FC and FS results of four samples. (**B**): WHC and FBC results of four samples. Different lowercase letters denote significant differences (*p* < 0.05).

**Table 1 foods-11-03960-t001:** Yield, solubility, basic components and color of four samples.

	FG	CPG	CG	BCG
Yield (%)	-	14.92 ± 0.68 ^c^	28.05 ± 0.47 ^a^	24.17 ± 0.96 ^b^
Solubility (%)	89.47 ± 0.26 ^b^	94.19 ± 1.09 ^a^	92.23 ± 0.38 ^ab^	92.07 ± 1.61 ^ab^
Protein (%)	86.81 ± 0.05 ^b^	93.61 ± 0.63 ^a^	92.55 ± 0.36 ^a^	92.45 ± 0.19 ^a^
Moisture (%)	8.82 ± 0.06 ^a^	4.35 ± 0.20 ^d^	4.83 ± 0.03 ^c^	5.33 ± 0.02 ^b^
Ash (%)	1.99 ± 0.08 ^a^	0.79 ± 0.04 ^b^	0.43 ± 0.02 ^c^	0.37 ± 0.00 ^c^
Fat (%)	0.13 ± 0.01 ^b^	0.13 ± 0.02 ^b^	0.39 ± 0.01 ^a^	0.14 ± 0.02 ^b^
L*	88.17 ± 0.17 ^b^	87.74 ± 0.81 ^b^	90.19 ± 0.23 ^a^	89.74 ± 0.23 ^a^
A*	1.21 ± 0.05 ^a^	–0.92 ± 0.07 ^b^	−1.05 ± 0.01 ^b^	−1.02 ± 0.02 ^b^
B*	14.38 ± 0.22 ^a^	6.91 ± 0.09 ^b^	5.23 ± 0.01 ^c^	5.47 ± 0.04 ^c^

FG in the table: Food-grade pigskin gelatin, CPG in the table: *Coregonus peled* scale gelatin, CG in the table: *Carp* scale gelatin, BCG in the table: *Bighead carp* scale gelatin. For the same row, different superscripts indicate significant differences (*p* < 0.05). L* lightness; A* redness and B* yellowness.

**Table 2 foods-11-03960-t002:** Amino acid composition (Amino acids g/100 g protein) of four samples.

Amino Acids	FG	CPG	CG	BCG
Asp	5.74	6.73	5.87	5.89
Thr	2.00	2.37	2.70	2.76
Ser	2.92	5.21	3.88	3.50
Glu	10.80	10.86	10.07	10.41
Gly	24.56	26.33	23.83	24.27
Ala	9.91	10.23	10.86	11.10
Val	2.43	2.02	2.15	2.04
Met	0.27	2.07	1.62	1.66
Ile	1.46	1.24	1.17	1.13
Leu	2.98	2.37	2.69	2.52
Tyr	0.21	0.19	0.26	0.26
Phe	2.06	2.22	2.19	2.27
Lys	3.91	3.77	3.77	3.90
His	0.71	1.39	0.71	0.55
Arg	8.61	9.11	8.91	9.08
Pro	11.96	9.59	10.50	11.05
Total	90.53	95.70	91.19	92.39

FG in the table: Food-grade pigskin gelatin, CPG in the table: *Coregonus peled* scale gelatin, CG in the table: *Carp* scale gelatin, BCG: *Bighead carp* scale gelatin.

**Table 3 foods-11-03960-t003:** Amide bond of four samples.

	FG	CPG	CG	BCG
Amide A	3294.22	3309.80	3305.47	3296.10
Amide B	3077.32	3080.68	3078.83	3080.05
Amide-I	1660.35	1663.11	1661.95	1657.27
Amide-II	1537.29	1547.43	1547.06	1542.57
Amide-III	1239.54	1239.90	1239.96	1240.55

FG in the table: Food-grade pigskin gelatin, CPG in the table: *Coregonus peled* scale gelatin, CG in the table: *Carp scale* gelatin, BCG in the table: *Bighead carp* scale gelatin.

**Table 4 foods-11-03960-t004:** Gel strength and TPA of four samples.

	FG	CPG	CG	BCG
Gel strength (g)	726.76 ± 10.01 ^a^	334.77 ± 11.31 ^c^	643.28 ± 8.42 ^b^	658.16 ± 5.85 ^b^
Hardness (g)	509.18 ± 15.18 ^a^	163.48 ± 15.85 ^c^	363.79 ± 7.22 ^b^	384.88 ± 5.13 ^b^
Springiness	0.95 ± 0.02 ^a^	0.94 ± 0.00 ^a^	0.95 ± 0.02 ^a^	0.95 ± 0.03 ^a^
Cohesiveness	0.90 ± 0.01 ^a^	0.71 ± 0.03 ^b^	0.91 ± 0.01 ^a^	0.85 ± 0.03 ^a^
Gumminess (g)	445.86 ± 12.72 ^a^	115.54 ± 9.45 ^c^	330.94 ± 6.84 ^b^	328.34 ± 5.23 ^b^
Chewiness (g)	424.81 ± 13.43 ^a^	109.10 ± 9.29 ^c^	312.62 ± 11.90 ^b^	311.84 ± 14.97 ^b^

FG in the table: Food-grade pigskin gelatin, CPG in the table: *Coregonus peled* scale gelatin, CG in the table: *Carp* scale gelatin, BCG in the table: *Bighead carp* scale gelatin. For the same row, different superscripts indicate significant differences (*p* < 0.05).

## Data Availability

The data presented in this study are available on request from the corresponding author.

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
