# Peer review of "The Structural and Functional Differences between Three Species of Fish Scale Gelatin and Pigskin Gelatin"

_foods, 2022, doi:10.3390/foods11243960_

Round 1

Reviewer 1 Report

This manuscript describes the structural and functional differences between three species of fish scale gelatin and pigskin gelatin. The results reported are of significance for application of fish scale gelatin. It is too extensive thus confusing and with lot of mistakes, missing some crucial discussion. Please find some specific comments below:

In lines 45, 47 and 48 please do not start the sentence with number of reference.  Start with ex. T. Zhang et al….

Line 78: Please explain equation 1, how did you determined S

Lines 97-103: Please explain how did you determine concentration in the supernatant

Lines 140 and 144: How did you choose concentration of gel solution to be exact 6.67 %

Line 179: Measuring volume­?

Line 209: Please rephrase in such way to indicate that FG higher ash content but it can be due to different extraction method.

Line 212-212: If FG was type B why you chose it for comparison with CPG, CG and BCG (they are type A)

Gelatin composition: Please discuss the results more, for example compare the obtained results with your previous paper “Effect of Phosphorylation on the Structure and Emulsification Properties of Different Fish Scale Gelatins” where you also had Coregonus pealed scale gelatin”. Some results are different but some are the same, so please explain. In addition, please explain the meaning of the parameters in Table 1 from the aspect of application. For example moisture and protein content are poorly discussed.

 Section 3.2.3.  Could you find the explanation for CPG good solubility even at pH 9, what is isoelectric point of CPG? Please discuss this.

Figure 1. Make Figure title shorter please, separate the figure like A and B in one figure and C and D second figure while E is separate third figure  since they are coming from different analyses. In addition after Discussion of figure 1 A and 1 B you are discussing SDS-PAGE which is given in Figure 2 and emulsifying properties are discussed much later (Figure 1C and 1D). It is confusing so please reorder the manuscript discussion and figures. The same comment stands for Figure 2 where different analyses are compressed in the same figure and discussed much later.

Line 284 Table 2 is not amino acid content.

Lines 315-317: Please provide some literature data for this explanation.

Lines 323-325: Can you make some connection of this result with some other results in your paper.

Line 359-361: Please discuss why.

Line 375: Table 2 is not zeta potential.

3.5. Please explain the smaller size of CPG emulsions in comparison to CG and BCG, and also discuss the small apposite value of zeta potential of FSG

In the Methods section 2.8.1, 2.8.1.1 and 2.8.1.2 are explained but there are no results and discussion in the paper.

Figure 3D is not discussed.

The conclusion does not correspond to the results in paper and it is focused just on the second part of the paper. Please prepare new conclusion with the most important outcomes.

Author Response

Cover Letter

Dear Editor and Reviewers,

With this letter we are submitting the revised version of our manuscript entitled ‘‘The structural and functional differences between three species of fish scale gelatin and pigskin gelatin’’ by Jinmeng He et al (Manuscript ID: foods-2023162). Your kind advices and those comments were all valuable and very helpful for revising and improving our paper, as well as the important guiding significance to our researches. Revised portion are marked in red in the paper. The responses to editors and reviewers’ comments/suggestions of our manuscript have been answered carefully point-by-point as follows. Some minor mistakes within the manuscript have also been corrected. We appreciate for Editors/Reviewers’ warm work earnestly, and hope that the correction will meet with approval.

Responses to the comments of Reviewer 1:

  1. In lines 45, 47 and 48 please do not start the sentence with number of reference. Start with ex. T. Zhang et al….

Response: Thanks for the reviewer's detailed suggestions. We have made modifications in Line 58, 60 and 61of revised manuscript.

  1. Line 78: Please explain equation 1, how did you determined S?

Response: We’re sorry to make you confused. In equation 1, S is the solid content of fish scale gelatin extract (%). In fact, when the extraction rate was calculated, the fish scale gelatin extract had been lyophilized. Therefore, S is 100%. Now I have changed equation 1 to E(%) = ×100%

  1. Lines 97-103: Please explain how did you determine concentration in the supernatant

Response: Thank the reviewers for their professional suggestions. The concentration of supernatant was determined by biuret method: 1mL supernatant was added with 4mLbiuret reagent, incubated at room temperature for 30min, then the absorbance at 540nm was measured by ultraviolet spectrophotometer, and the concentration was calculated by the standard curve made of bovine serum albumin.

  1. Lines 140 and 144: How did you choose concentration of gel solution to be exact 6.67 %?

Response: We’re sorry to make you confused. At present, the method described by Bloom (O.T. Bloom, United States Patent No. 1,540,979, U.S. Patent and Trademark, Washington, DC, 1925) is commonly used to study the gel strength of gelatin, and the gelatin concentration of the method is 6.67%, so this concentration is also selected in this study.

  1. Line 179: Measuring volume­?

Response: We’re sorry to make you confused. When measuring the foaming property of gelatin, the gelatin solution is packed in a 50ml plastic centrifuge tube with scale, so the volume of solution and foam can be directly measured.

  1. Line 209: Please rephrase in such way to indicate that FG higher ash content but it can be due to different extraction method.

Response: Thanks for the reviewer's detailed suggestions. We have made modifications in Line 266-267 of revised manuscript.

  1. Line 212-212: If FG was type B why you chose it for comparison with CPG, CG and BCG (they are type A)

Response: We’re sorry to make you confused. The purpose of this article is to compare the differences in structure and function between FSGs and FG, and to explore the possibility of using FSG instead of FG. Acid method is used to prepare FSGs because inorganic salts such as hydroxyapatite in fish scale structure are removed. Considering the structure of pigskin and its economic benefits in actual production, it is a common method to prepare pigskin gelatin by alkali method. Therefore, type B FG and type A FSG are selected for comparison.

  1. Gelatin composition: Please discuss the results more, for example compare the obtained results with your previous paper “Effect of Phosphorylation on the Structure and Emulsification Properties of Different Fish Scale Gelatins” where you also had Coregonus pealed scale gelatin”. Some results are different but some are the same, so please explain. In addition, please explain the meaning of the parameters in Table 1 from the aspect of application. For example moisture and protein content are poorly discussed.

Response: We’re sorry to make you confused. In this study and previous studies, there are some differences in the extraction process of gelatin from fish scales, such as the ratio of fish scales to hydrochloric acid solution, which is 1: 10 in this paper, while the other one is 1:15. After acid treatment, it is necessary to clean the residual hydrochloric acid, which may easily cause the loss of fish scales. And it is easy to cause losses in the process of transfer and filtration after fish scale extraction and centrifugation. In addition, the fish scales are different in different harvest seasons, etc. These details may be the reasons for the differences in extraction rate and fat of gelatin extracted by us. The parameters in Table 1 are mainly used to analyze whether the extracted three kinds of FSG can meet the application standards in food. Some discussions on protein and moisture content are added in Line 256-264.

  1. Section 3.2.3. Could you find the explanation for CPG good solubility even at pH 9, what is isoelectric point of CPG? Please discuss this.

Response: We’re sorry to make you confused. We have made modifications in Line 301-310 of revised manuscript. When Zhang et al. studied the turbidity of commercial cold-water fish skin gelatin, they found that it has good solubility at any pH, which means that commercial cold-water fish skin gelatin has no obvious isoelectric point. However, there is no clear explanation for this result. The research of Tamnak et al. thinks that the low-content folding structure is beneficial to contact with hydrophilic molecules, so that the protein has good solubility, and the research of Feng et al. thinks that the solubility of gelatin is related to the composition of its subunits, and the low molecular weight subunits are more beneficial to the solubility of protein. Among the four kinds of gelatin, CPG has the lowest content of β -sheet structure (Fig.3B) SDS-PAGE (Fig.2) shows that CPG has more subunits with lower molecular weight distribution than the other three kinds of gelatin, which may be the reason why CPG shows better solubility at all pH values. For every protein, there is a pH that makes its surface net charge zero, and the pH at this time is the isoelectric point (pI). When the potential of the emulsion was measured, the emulsion prepared by CPG was positively charged at pH9, which indicated that CPG’s pI > 9.

Reference

Zhang, T.; Sun, R.; Ding, M.; Li, L.; Tao, N.; Wang, X.; Zhong, J. Commercial cold-water fish skin gelatin and bovine bone gelatin: Structural, functional, and emulsion stability differences. Lwt 2020, 125, doi:10.1016/j.lwt.2020.109207.

Tamnak, S.; Mirhosseini, H.; Tan, C.P.; Ghazali, H.M.; Muhammad, K. Physicochemical properties, rheological behavior and morphology of pectin-pea protein isolate mixtures and conjugates in aqueous system and oil in water emulsion. Food Hydrocolloids 2016, 56, 405-416, doi:10.1016/j.foodhyd.2015.12.033.

Feng, X.; Dai, H.; Zhu, J.; Ma, L.; Yu, Y.; Zhu, H.; Wang, H.; Sun, Y.; Tan, H.; Zhang, Y. Improved solubility and interface properties of pigskin gelatin by microwave irradiation. Int J Biol Macromol 2021, 171, 1-9, doi:10.1016/j.ijbiomac.2020.12.215.

  1. Figure 1. Make Figure title shorter please, separate the figure like A and B in one figure and C and D second figure while E is separate third figure since they are coming from different analyses. In addition after Discussion of figure 1 A and 1 B you are discussing SDS-PAGE which is given in Figure 2 and emulsifying properties are discussed much later (Figure 1C and 1D). It is confusing so please reorder the manuscript discussion and figures. The same comment stands for Figure 2 where different analyses are compressed in the same figure and discussed much later.

Response: Thanks for the reviewer's detailed suggestions. We have made modifications in Line 336-337 of revised manuscript. Figure. 1, Figure. 2 and Figure. 3 have been reordered as follows: Figure. 1A, Figure. 1B, Figure. 1C, Figure. 1D and Figure. 1E have been changed to Figure. 1A, Figure. 1B, Figure. 6A, Figure. 6B and Figure. 5B, respectively; Figure. 2A, Figure. 2B, Figure.2C, Figure. 2D and Figure. 2E have been changed to Figure. 2, Figure. 3A, Figure. 4A, Figure.3B and Figure. 4B, respectively; Figure. 3A, Figure. 3B, Figure.3C, and Figure. 3D have been changed to Figure.5A, Figure. 7A, Figure. 7B and Figure. 5C, respectively. And these pictures are placed near the discussion paragraph.

  1. Line 284 Table 2 is not amino acid content.

Response: Thank you so much for your careful check. Table 2 is the composition of four gelatin amino acids, but it was not placed near this analysis. We have adjusted the position of the table to Line 361-363 of revised manuscript.

  1. Lines 315-317: Please provide some literature data for this explanation.

Response: Thanks for the reviewer's detailed suggestions. After our examination, we found that this sentence was incorrectly expressed. We have made modifications in Line 401-406 of revised manuscript. Detailed revisions are as following:

The peak positions of four kinds of gelatin are all at 290nm, but their fluorescence intensities are different, with the following magnitude: BCG > CG > FG > CPG. The willingness to cause this difference may be due to their different tertiary structures. The higher fluorescence intensity of BCG may be due to the fact that more aromatic groups are exposed to water and it is easier to emit fluorescence. The low fluorescence intensity of CPG may be due to the aggregation of some nonpolar amino acid residues covered by protein.

Reference

Xu, X.; Liu, W.; Liu, C.; Luo, L.; Chen, J.; Luo, S.; McClements, D.J.; Wu, L. Effect of limited enzymatic hydrolysis on structure and emulsifying properties of rice glutelin. Food Hydrocolloids 2016, 61, 251-260, doi:10.1016/j.foodhyd.2016.05.023.

  1. Lines 323-325: Can you make some connection of this result with some other results in your paper.

Response: Thanks for the reviewer's detailed suggestions. We have made modifications in Line 414-423 of revised manuscript. Detailed revisions are as following:

 It was found that cold-water fish usually showed higher hydrophobicity, while warm-water fish contained higher proline content and showed good hydrophilicity. Similar to the results of this paper, coregonus peled is a kind of cold-water fish, and the gelatin extracted from it has the lowest proline content among the four kinds of gelatin. In addition, the aggregation of protein will reduce the surface area of hydrophobic groups exposed to water, resulting in the decrease of surface hydrophobicity. It can be found from Figures 1A and b that the absorbance of CPG is the lowest at pH7, indicating that the aggregation degree of protein in this solution is low. This may be another reason for its high surface hydrophobicity.

Reference

Lv, L.-C.; Huang, Q.-Y.; Ding, W.; Xiao, X.-H.; Zhang, H.-Y.; Xiong, L.-X. Fish gelatin: The novel potential applications. Journal of Functional Foods 2019, 63, doi:10.1016/j.jff.2019.103581.

Xu, X.; Liu, W.; Liu, C.; Luo, L.; Chen, J.; Luo, S.; McClements, D.J.; Wu, L. Effect of limited enzymatic hydrolysis on structure and emulsifying properties of rice glutelin. Food Hydrocolloids 2016, 61, 251-260, doi:10.1016/j.foodhyd.2016.05.023.

  1. Line 359-361: Please discuss why.

Response: Thanks for the reviewer's detailed suggestions. We have made modifications in Line 475-483 of revised manuscript. Detailed revisions are as following:

The particle size of the four kinds of gelatin follows: BCG (653.30 nm) ≈ CG (627.70 nm) > CPG (552.95 nm) > FG (428.00 nm) (Fig.6A). Huang et al.' s research shows that the smaller the absolute value of charge may lead to less electrostatic repulsion between droplets, thus promoting droplet flocculation to form large particle size. As shown in Figure 6B, the absolute value of the ζ-potential follows: FG (-25.10 mV) > CPG (3.03 mV) > CG (-1.53 mV) ≈ BCG (-1.51 mV). Therefore, FG emulsion has a smaller particle size because it has a higher negative charge and can resist the aggregation of droplets. On the contrary, BCG and CG have the lowest absolute charge, which makes their aggregation resistance the worst and their particle size the largest.

Reference

Huang, T.; Tu, Z.C.; Shangguan, X.; Wang, H.; Sha, X.; Bansal, N. Rheological behavior, emulsifying properties and structural characterization of phosphorylated fish gelatin. Food Chem 2018, 246, 428-436, doi:10.1016/j.foodchem.2017.12.023.

  1. Line 375: Table 2 is not zeta potential.

Response: Thank you so much for your careful check. It is our negligence and we are sorry about this. Table 2 has been changed to Figure 6B in Line 483.

  1. 5. Please explain the smaller size of CPG emulsions in comparison to CG and BCG, and also discuss the small apposite value of zeta potential of FSG

Response: Thanks a lot for your excellent advice. Among the three FSFs, the absolute value of ζ-potential of CPG is 3.03, while CG and BCG are 1.53 and 1.51, respectively. Therefore, CPG has greater electrostatic repulsion force, which inhibits emulsion flocculation to form large particles, resulting in smaller particle size of CPG emulsion than CG and BCG emulsion. FSGs is a type A gelatin prepared by acid method. Considering the turbidity, the isoelectric point of BCG and CG is about pH 9. Therefore, the emulsion prepared by them at pH 9 has a small potential. The range of FSGs potential needs to be further analyzed by measuring the potential at different pH values.

Reference

Huang, T.; Tu, Z.C.; Shangguan, X.; Wang, H.; Sha, X.; Bansal, N. Rheological behavior, emulsifying properties and structural characterization of phosphorylated fish gelatin. Food Chem 2018, 246, 428-436, doi:10.1016/j.foodchem.2017.12.023.

  1. In the Methods section 2.8.1, 2.8.1.1 and 2.8.1.2 are explained but there are no results and discussion in the paper.

Response: Thank you so much for your careful check. We discussed Section 2.8.1, 2.8.1.1, and 2.8.1.2 in Line 426-445.

  1. Figure 3D is not discussed.

Response: Thank you so much for your careful check. Figure 3D is adjusted to Figure 5C, and it is discussed in Line 506-510.

  1. The conclusion does not correspond to the results in paper and it is focused just on the second part of the paper. Please prepare new conclusion with the most important outcomes.

Response: Thanks for the reviewer's good evaluation and detailed suggestions. We have made modifications in Line 546-561 of revised manuscript.

To sum up, we have rechecked the full text and try our best to revise it on the basis of comments. We hope this revised manuscript has addressed your concerns. Please do not hesitate to contact us at the address below if there are any questions.

Tel: +86-15228394850

Email: [email protected].

Thank you again for your hard work!

Best wishes to you!

 Yours sincerely,

Jinmeng He

2022-11-15

Reviewer 2 Report

This Ms examines the structural and functional differences between the three FSGs and FG, and the results suggest that many functional properties of FG are still better choices. It is difficult to conclude from the current data that FSGs have more commercial potential, and rather require some revision. See my comments below.

1)       Abstract, Authors previously stated that FG is superior to FSGs in many functional properties. Why do the authors still conclude that FSGs has more potential?

2)       Line 30-34, These sentences should be rephrased in a more concise form.

3)       Line 42, The study of food science is not suitable to discuss customs, religion, etc.

4)       The Ms suffers from many formatting The manuscript suffers from many formatting problems, such as the citation format in lines 127 and 132, and the use of percentage signs, such as in lines 174, and 181-182. Authors should check and revise carefully.

5)       Line 300-307, This needs to be discussed in depth instead of just stating the results. What is the purpose of the authors doing this experiment?

6)       Line 408-410, This contradicts the previous conclusion in the Ms.

7)       The conclusion of the Ms is not complete. The Ms studied the comparison of three FSGs and FG, but the authors only mentioned the comparison of two FSGs (CG, BCG) and FG in the final conclusion, and did not completely analyze the four gelatins.

Author Response

Cover Letter

Dear Editor and Reviewers,

With this letter we are submitting the revised version of our manuscript entitled ‘‘The structural and functional differences between three species of fish scale gelatin and pigskin gelatin’’ by Jinmeng He et al (Manuscript ID: foods-2023162). Your kind advices and those comments were all valuable and very helpful for revising and improving our paper, as well as the important guiding significance to our researches. Revised portion are marked in red in the paper. The responses to editors and reviewers’ comments/suggestions of our manuscript have been answered carefully point-by-point as follows. Some minor mistakes within the manuscript have also been corrected. We appreciate for Editors/Reviewers’ warm work earnestly, and hope that the correction will meet with approval.

Responses to the comments of Reviewer 2:

  1. Abstract, Authors previously stated that FG is superior to FSGs in many functional properties. Why do the authors still conclude that FSGs has more potential?

Response: We’re sorry to make you confused. It is our negligence and we are sorry about this. What we want to express is that FSGs can replace FG and has the possibility of industrial application. The modification has been made in Line 8-20 of revised manuscript.

  1. Line 30-34, These sentences should be rephrased in a more concise form.

Response: Thanks for the reviewer's detailed suggestions. We have made modifications in Line 43-45 of revised manuscript.

  1. Line 42, The study of food science is not suitable to discuss customs, religion, etc.

Response: Thanks for the reviewer's detailed suggestions. We have made modifications in Line 55-56 of revised manuscript.

  1. The Ms suffers from many formatting the manuscript suffers from many formatting problems, such as the citation format in lines 127 and 132, and the use of percentage signs, such as in lines 174, and 181-182. Authors should check and revise carefully.

Response: Thank you so much for your careful check. The modification has been made in Line 158&165&213&231-232 of revised manuscript.

  1. Line 300-307, This needs to be discussed in depth instead of just stating the results. What is the purpose of the authors doing this experiment?

Response: We’re sorry to make you confused. We did this experiment to analyze the interaction between groups in the secondary structure of gelatin. As the reviewer’s advice, we have made modifications in Line 365-379 of revised manuscript.

  1. Line 408-410, This contradicts the previous conclusion in the Ms.

Response: Thanks a lot for your reminding. We have made modifications in Line 518-527 of revised manuscript.

  1. The conclusion of the Ms is not complete. The Ms studied the comparison of three FSGs and FG, but the authors only mentioned the comparison of two FSGs (CG, BCG) and FG in the final conclusion, and did not completely analyze the four gelatins.

Response: Thanks for your kind reminding. We have made modifications in Line 546-561 of revised manuscript.

To sum up, we have rechecked the full text and try our best to revise it on the basis of comments. We hope this revised manuscript has addressed your concerns. Please do not hesitate to contact us at the address below if there are any questions.

Tel: +86-15228394850

Email: [email protected].

Thank you again for your hard work!

Best wishes to you!

 Yours sincerely,

Jinmeng He

2022-11-15

Reviewer 3 Report

The current manuscript describes the difference between structural and functional properties of fish scale gelation and food grade pigskin gelation. Some interesting data’s are presented in the manuscript. Final results of the manuscript suggest that fish scale gelation have commercial development potential. However, author missed few points that drags down this manuscript.

 ·      Highlight the significant outcome in abstract

 ·      Introduction is too short, explain in more detail

 ·      Materials and method: The equation used in manuscript is not clear and   symbols are not addressed correctly. For example, A0 instead of A0, C & c is not differentiated properly in equation. Please rewrite all equation using π equations or formula writing.

 ·    Table numbers are wrongly marked and alignment is not good (for tables), please change it 

 ·     Figure 2A is not clear, add the clear picture

 ·     In figures, x axis title is missing, add the axis title to all figures presented in manuscript

 ·    Add the future perspectives and significant outcome in conclusion part, that establish your work.

Author Response

Cover Letter

Dear Editor and Reviewers,

With this letter we are submitting the revised version of our manuscript entitled ‘‘The structural and functional differences between three species of fish scale gelatin and pigskin gelatin’’ by Jinmeng He et al (Manuscript ID: foods-2023162). Your kind advices and those comments were all valuable and very helpful for revising and improving our paper, as well as the important guiding significance to our researches. Revised portion are marked in red in the paper. The responses to editors and reviewers’ comments/suggestions of our manuscript have been answered carefully point-by-point as follows. Some minor mistakes within the manuscript have also been corrected. We appreciate for Editors/Reviewers’ warm work earnestly, and hope that the correction will meet with approval.

Responses to the comments of Reviewer 3:

  1. Highlight the significant outcome in abstract

Response: Thank you for your constructive comment. The modification has been made in Line 8-20 of revised manuscript.

  1. Introduction is too short, explain in more detail

Response: Thank you very much for your thoughtful advice and comments. We added some contents in Lines 27-33&37-45&46-50&62-66 of revised manuscript.

  1. Materials and method: The equation used in manuscript is not clear and symbols are not addressed correctly. For example, A0 instead of A0, C & c is not differentiated properly in equation. Please rewrite all equation using π equations or formula writing.

Response: Thanks a lot for the very careful comments and apologize for the mistakes caused by our carelessness. We rewrote all equation using π equations in Line 95&126&205-206&223&231-232&242 of revised manuscript.

  1. Table numbers are wrongly marked and alignment is not good (for tables), please change it 

Response: Thanks a lot for the very careful comments and apologize for the mistakes caused by our carelessness. In the revised manuscript, the table number and alignment of have been revised in Line 272&361&380&466.

  1. Figure 2A is not clear, add the clear picture

Response: Thanks a lot for your reminding. The modification has been made in Line 339-241 of revised manuscript.

  1. In figures, x axis title is missing, add the axis title to all figures presented in manuscript

Response: Thank you so much for your careful check and apologize for the mistakes caused by our carelessness. we added the axis title to all figures of revised manuscript.

  1. Add the future perspectives and significant outcome in conclusion part, that establish your work.

 Response: Thanks a lot for your excellent advice. The modification has been made in Line 556-561 of revised manuscript.

To sum up, we have rechecked the full text and try our best to revise it on the basis of comments. We hope this revised manuscript has addressed your concerns. Please do not hesitate to contact us at the address below if there are any questions.

Tel: +86-15228394850

Email: [email protected].

Thank you again for your hard work!

Best wishes to you!

 Yours sincerely,

Jinmeng He

2022-11-15

Reviewer 4 Report

Comments for authors

The manuscript entitled “The structural and functional differences between three species of fish scale gelatin and pigskin gelatin” is informative and well-studied.

General comments:

Line 8-16: Rewrite the abstract. The abstract need to have the numerical values of the various properties (gel strength, emulsifying properties, foaming properties and water-holding capacity) tested. The abstract should be concise and clear.

Line 23-24: Provide the recent data about gelatin production at global level.

Line 27-30: Rewrite the statement. The statement is not clearly presented.

Line 38-39: Replace the word pressing with another appropriate word. Mention either fishing industry or fishery industry in the paper, keep it uniform.

Line 39-41: The sentence is grammatically wrong, rewrite it.

Line 52: Replace the word emulsification with emulsifying.

Line 53 and 55: replace the word between with among.

Line 70: Provide the technical details mechanical stirrer with the rotation speed and time.

Line 75: Mention the technical details of centrifugation process with relevant parameters used for the process.

Line 76: It should be clothes not cloths.

Line 88: Provide the technical details of colorimeter.

Line 89-91: The sentence is poorly written and incomplete. Rephrase the whole statement.

Line 95-96: Mention the technical details of digital camera and UV-vis spectrophotometer.

Line 98: r/m is not the correct way of writing. Replace it by rpm.

Line 102-103: Mention the units of different parameters in the formula.

Line 106: mention the full form of SDS-PAGE.

Section 2.7.1: The method is not clearly stated. The technical details of the equipment's are missing, and the choice of words are not correct and not depicting the meaning clearly.

Line 116 and 117: Rewrite the whole sentences. It is poorly written.

Line 118-120: The statement is technically weak. What does vacuum mean?

Section 2.7.3: Please provide the reference for the FTIR process. Also do mention the clear version of the software.

Section 2.7.4 and 2.7.5: Describe the process clearly and make it grammatically appropriate.

Line 135-137: Merge the sentence in the respective sections of analysis. It is meaningless to mention in separate subsections.

Line 140: Rewrite the sentence. Fix the grammatical errors.

Line 152-155: fix the grammatical errors in the whole statement.

Line 157 and 158: mention “o” in subscript with “A”.

Section 2.8.2.2: Rewrite the whole section and fix the grammatical errors.

Line 170: Replace d with day.

Line 179: What does measuring value mean?

Line 190: Mention the mesh size of filter paper.

Line 222-224: The results are not clearly presented and a proper citations should be made in reference to some previous studies justifying the results presented.

Line 245: Add ‘are in the ………… range of…………..’

Fig. 1C and 1E: Provide the X-axis titles.

Fig. 1B: Indicate error bar and alphabetical notation. Further the discussion is missing for absorbance of the various gelatins.

Line 285: The word content is missing.

Section 3.3.2: The discussion is incomplete and not precise. The discussion should include information about the amino acid compositions of all gelatins being studied in the reserch.

Line 300: TableS1 is missing in the paper.

Line 315-317: The discussion is not clearly mentioned? What is the reason behind the decreasing fluorescence intensity of gelatins in the order mentioned?

Line 321: Mention the full form of ANS.

Line 332: Replace imino with amino.

Section 3.4 The discussion lacks precise discussion. Provide some proper citation and references to justify the results obtained in the study.

Line 384-387: Provide suitable reference for the statement.

Section 3.6: The paragraph lacks proper justification. Provide more references and precise justification for the result obtained.

Line 418-419: Be very careful while writing the sentences for grammar and formatting issues. Rewrite it.

Section 4: Be careful of grammar issues. Rewrite the conclusion.

Overall comments

1.     Keep the values after decimal places uniform throughout the paper.

2.     Strictly follow journal’s guidelines while writing “Figure or Table” within the results and discussion section, moreover in the entire paper.

3.     Maintain a uniform format for all the references as per journal’s guidelines.

Author Response

Cover Letter

Dear Editor and Reviewers,

With this letter we are submitting the revised version of our manuscript entitled ‘‘The structural and functional differences between three species of fish scale gelatin and pigskin gelatin’’ by Jinmeng He et al (Manuscript ID: foods-2023162). Your kind advices and those comments were all valuable and very helpful for revising and improving our paper, as well as the important guiding significance to our researches. Revised portion are marked in red in the paper. The responses to editors and reviewers’ comments/suggestions of our manuscript have been answered carefully point-by-point as follows. Some minor mistakes within the manuscript have also been corrected. We appreciate for Editors/Reviewers’ warm work earnestly, and hope that the correction will meet with approval.

Responses to the comments of Reviewer 4:

  1. Line 8-16: Rewrite the abstract. The abstract need to have the numerical values of the various properties (gel strength, emulsifying properties, foaming properties and water-holding capacity) tested. The abstract should be concise and clear.

Response: Thank you for your constructive comment. The modification has been made in Line 8-20 of revised manuscript.

  1. Line 23-24: Provide the recent data about gelatin production at global level.

Response: Thank you for your good suggestions. The modification has been made in Line 27-33 of revised manuscript.

  1. Line 27-30: Rewrite the statement. The statement is not clearly presented.

Response: Thanks a lot for your excellent advice. The modification has been made in Line 36-39 of revised manuscript. Detailed revisions are as following:

However, the application of mammalian gelatin is limited by consumer concerns about the spread of bovine spongiform encephalopathy, foot-and-mouth disease, hand-foot-mouth disease and other diseases, and some special dietary needs, the application of mammalian gelatin is limited.

  1. Line 38-39: Replace the word pressing with another appropriate word. Mention either fishing industry or fishery industry in the paper, keep it uniform.

Response: Thanks for the reviewer's detailed suggestions. The modification has been made in Line 46&52 of revised manuscript. The word pressing has been changed to urgent, and fishing industry has been changed to fishery industry.

  1. Line 39-41: The sentence is grammatically wrong, rewrite it.

Response: Thank you so much for your careful check. The modification has been made in Line 53-54 of revised manuscript. Detailed revisions are as following:

 Fish scale is one of the wastes produced in the process of fish processing, which accounts for about 3% of the total weight of fish and is rich in collagen.

  1. Line 52: Replace the word emulsification with emulsifying.

Response: Thanks for the reviewer's detailed suggestions. In revised manuscript, the word emulsification has been changed to emulsifying.

  1. Line 53 and 55: replace the word between with among.

Response: Thanks for the reviewer's detailed suggestions. The modification has been made in Line 70&72 of revised manuscript.

  1. Line 70: Provide the technical details mechanical stirrer with the rotation speed and time.

Response: Thanks for the reviewer's detailed suggestions. The modification has been made in Line 87 of revised manuscript. The technical details of mechanical stirrer have been supplemented, with the rotation speed of 1000rpm and the time of 30min.

  1. Line 75: Mention the technical details of centrifugation process with relevant parameters used for the process.

Response: Thanks for the reviewer's detailed suggestions. The modification has been made in Line 92 of revised manuscript. The technical details of centrifugation process have been supplemented, with the rotation speed of 5000rpm and the time of 20 min.

  1. Line 76: It should be clothes not cloths.

Response: Thanks a lot for the very careful comment and apologize for the mistake caused by our carelessness. In revised manuscript, the word cloths have been changed to clothes in Line 93.

  1. Line 88: Provide the technical details of colorimeter.

Response: Thanks for the reviewer's detailed suggestions. In revised manuscript, we provided the technical details of colorimeter in Line 105-106.

  1. Line 89-91: The sentence is poorly written and incomplete. Rephrase the whole statement.

Response: Thanks a lot for the very careful comment and apologize for the mistake caused by our carelessness. The modification has been made in Line 107-108 of revised manuscript.

  1. Line 95-96: Mention the technical details of digital camera and UV-vis spectrophotometer.

Response: Thanks for the reviewer's detailed suggestions. In revised manuscript, we provided the technical details of colorimeter in Line 117-121.

  1. Line 98: r/m is not the correct way of writing. Replace it by rpm.

Response: Thank you so much for your careful check. The modification has been made in Line 123 of revised manuscript.

  1. Line 102-103: Mention the units of different parameters in the formula.

Response: Thanks for the reviewer's detailed suggestions. In revised manuscript, we added the units of different parameters in the formula in Line 127-128.

  1. Line 106: mention the full form of SDS-PAGE.

Response: Thanks for the reviewer's detailed suggestions. In revised manuscript, we added the full form of SDS-PAGE in Line 130.

  1. Section 2.7.1: The method is not clearly stated. The technical details of the equipment's are missing, and the choice of words are not correct and not depicting the meaning clearly.

Response: Thanks a lot for the very careful comment and apologize for the mistake caused by our carelessness. The modification has been made in Line 132&134-137&139 of revised manuscript.

  1. Line 116 and 117: Rewrite the whole sentences. It is poorly written.

Response: Thanks for the reviewer's detailed suggestions. In revised manuscript, this sentence has been rewritten in Line 142-143.

  1. Line 118-120: The statement is technically weak. What does vacuum mean?

Response: Thanks for the reviewer's detailed suggestions. In revised manuscript, some technical details have been added in Line 143-148. Detailed revisions are as following: after hydrolysis, the sample was filtered into a 50ml volumetric flask, and after constant volume, 1mL was transferred into a test tube, which was blown to dryness with nitrogen, add 1 mL of ultrapure water into the test tube, blow dry with nitrogen, then dissolved in 1 mL of 0.2 mol/L sodium citrate buffer, filtered through a 0.22μm filter membrane, and injected into an amino acid analyzer (L-8900, Hitachi, Tokyo, Japan) for determination. Vacuum is an inaccurate description. Now I've changed it to blown to dryness with nitrogen.

  1. Section 2.7.3: Please provide the reference for the FTIR process. Also do mention the clear version of the software.

Response: Thanks for the reviewer's detailed suggestions. The modification has been made in Line 151&155-156 of revised manuscript.

The reference for the FTIR process of samples for is as follow:

Yang, M.; Zhang, J.; Guo, X.; Deng, X.; Kang, S.; Zhu, X.; Guo, X. Effect of Phosphorylation on the Structure and Emulsification Properties of Different Fish Scale Gelatins. Foods 2022, 11, doi:10.3390/foods11060804.

  1. Section 2.7.4 and 2.7.5: Describe the process clearly and make it grammatically appropriate.

Response: Thanks for the reviewer's detailed suggestions. The modification has been made in Line 159-163&166-178 of revised manuscript.

The process in Section 2.7.4 has been added: Gelatins were dissolved in 10 mmol/L phosphate buffer solution with a pH of 7.0, and the test sample solution containing 0.2 mg/mL gelatin was prepared. The fluorescence spectrometer (S8 TIFER, Bruker, Germany) was operated with an excitation wavelength of 295 nm and a slit width of 10 nm. The fluorescence emission spectra were collected in the range of 260–350 nm.

The process in Section 2.7.5 has been added: The surface hydrophobicity index of gelatin was determined by using ammonium salt of 1-anilino -8- naphthalenesulfonate (ANS) as fluorescent probe. Gelatin was prepared into solutions with different concentrations (0.02mg/ml, 0.05mg/ml, 0.1mg/ml, 0.2mg/ml, 0.5mg/ml, respectively) Make ANS into 8mM solution with the same phosphate buffer solution, and keep it away from light for later use. 40μL ANS solution and 4mL gelatin solution were mixed evenly, and the fluorescence intensity of the mixed solution was detected by fluorescence spectrometer (S8 TIFER, Bruker, Germany). Parameters: excitation wavelength 390nm, emission wavelength 470nm, slit width 10nm, and test temperature 25℃. The difference between the fluorescence intensity of gelatin solution with ANS and that without ANS is the relative fluorescence intensity. The relative fluorescence intensity of gelatin solution is plotted against the intensity of gelatin solution, and the initial slope of the curve is the surface hydrophobicity index of gelatin to be measured.

  1. Line 135-137: Merge the sentence in the respective sections of analysis. It is meaningless to mention in separate subsections.

Response: Thanks for the reviewer's detailed suggestions. In revised manuscript, this sentence has been merged into 2.8.1.1 and 2.8.1.2 in Line 184-186&191-192.

  1. Line 140: Rewrite the sentence. Fix the grammatical errors.

Response: Thank you so much for your careful check. The modification has been made in Line 183-184 of revised manuscript.

  1. Line 152-155: fix the grammatical errors in the whole statement.

Response: Thank you so much for your careful check. The modification has been made in Line 198-202 of revised manuscript.

  1. Line 157 and 158: mention “o” in subscript with “A”.

Response: Thanks for the reviewer's detailed suggestions. In revised manuscript, the A0 has been changed to A0 in Line 205-206.

  1. Section 2.8.2.2: Rewrite the whole section and fix the grammatical errors.

Response: Thanks for the reviewer's detailed suggestions. The modification has been made in Line 211-216 of revised manuscript.

  1. Line 170: Replace d with day.

Response: Thanks for the reviewer's detailed suggestions. The modification has been made in Line 219 of revised manuscript.

  1. Line 179: What does measuring value mean?

Response: We’re sorry to make you confused. Gelatin solution is placed in a 50ml plastic centrifuge tube with scale, so the volume of solution and foam can be directly read after homogenization. The measured volume refers to the total volume of solution and foam. In revised manuscript, we added some details in Line 228-229.

  1. Line 190: Mention the mesh size of filter paper.

Response: Thanks for the reviewer's detailed suggestions. In revised manuscript, we added mesh size of filter paper in Line 240.

  1. Line 222-224: The results are not clearly presented and a proper citations should be made in reference to some previous studies justifying the results presented.

Response: Thank you for your constructive comment. In revised manuscript, we cited some references to demonstrate the results in Line 280-281. Detailed revisions are as following:

 The lightness (L*) value of four gelatin samples was in the range of 87.74–90.19, and CG had the highest L* value. The L* value from European eel skin gelatin was 89.28, which was also in this range, and it’s a* and b* were -0.12 and 11.70, respectively.

References

Sila, A.; Martinez-Alvarez, O.; Krichen, F.; Gómez-Guillén, M.C.; Bougatef, A. Gelatin prepared from European eel (Anguilla anguilla) skin: Physicochemical, textural, viscoelastic and surface properties. Colloids and Surfaces A: Physicochemical and Engineering Aspects 2017, 529, 643-650, doi:10.1016/j.colsurfa.2017.06.032.

  1. Line 245: Add ‘are in the ………… range of…………..’

Response: Thanks for the reviewer's detailed suggestions. The modification has been made in Line 314 of revised manuscript.

  1. 1C and 1E: Provide the X-axis titles.

Response: Thanks for the reviewer's detailed suggestions. In revised manuscript, Fig.1C and Fig.1E are changed to Figure 6A and Figure 5B, and X-axis titles are added.

  1. 1B: Indicate error bar and alphabetical notation. Further the discussion is missing for absorbance of the various gelatins.

Response: Thank you very much for your thoughtful advice. In revised manuscript, the error bars and alphabetical notation of Fig.1B have been supplemented. In addition, the discussion on the absorbance of various gelatins is supplemented in Line 301-312.

  1. Line 285: The word content is missing.

Response: Thank you so much for your careful check. The modification has been made in Line 349 of revised manuscript.

  1. Section 3.3.2: The discussion is incomplete and not precise. The discussion should include information about the amino acid compositions of all gelatins being studied in the reserch.

Response: Thank you very much for your thoughtful advice. In revised manuscript, our discussion on amino acid composition is supplemented in Line 346-354&256-360.

  1. Line 300: Table S1 is missing in the paper.

Response: Thanks a lot for the very careful comments and apologize for the mistakes caused by our carelessness. In revised manuscript, Table S1 has been supplemented as Table 3 in Line 380-382.

  1. Line 315-317: The discussion is not clearly mentioned? What is the reason behind the decreasing fluorescence intensity of gelatins in the order mentioned?

Response: We’re sorry to make you confused. The reasons for the difference in fluorescence intensity of four kinds of gelatin are analyzed in Line 346-354&356-360 of revised manuscript. Detailed revisions are as following:

The fluorescence spectrum of gelatin is shown in Fig. 4A. The peak positions of four kinds of gelatin are all at 290nm, but their fluorescence intensities are different, with the following magnitude: BCG > CG > FG > CPG. The willingness to cause this difference may be due to their different tertiary structures. The higher fluorescence intensity of BCG may be due to the fact that more aromatic groups are exposed to water and it is easier to emit fluorescence. The low fluorescence intensity of CPG may be due to the aggregation of some nonpolar amino acid residues covered by protein.

References

Xu, X.; Liu, W.; Liu, C.; Luo, L.; Chen, J.; Luo, S.; McClements, D.J.; Wu, L. Effect of limited enzymatic hydrolysis on structure and emulsifying properties of rice glutelin. Food Hydrocolloids 2016, 61, 251-260, doi:10.1016/j.foodhyd.2016.05.023.

  1. Line 321: Mention the full form of ANS.

Response: Thank you so much for your careful check. The modification has been made in Line 167 of revised manuscript.

  1. Line 332: Replace imino with amino.

Response: Thank you so much for your detailed suggestions. The modification has been made in Line 431 of revised manuscript.

  1. Section 3.4 The discussion lacks precise discussion. Provide some proper citation and references to justify the results obtained in the study.

Response: Thanks for the reviewer's good evaluation and detailed suggestions. As the reviewer’s advice, we have made modifications in Line 436-445 of revised manuscript. Detailed revisions are as following:

Springiness can be used to characterize the rubber-like elasticity of colloid in mouth and the influence of initial deformation on colloid structure. In this study, the springiness of four kinds of gelatin has similar results. Cohesiveness can be used to measure the integrity of the network structure of the gel after extrusion, and can be used for comparative analysis of the influence of the first extrusion and the second extrusion on the gel structure. Among the four kinds of gelatin, CPG has the lowest cohesiveness, which indicates that the network structure of the gel is destroyed most seriously after extrusion, and chewiness can represent the taste of human mouth when chewing. Generally, chewiness is consistent with the change trend of gumminess and hardness of the gel, which is consistent with the results of this study.

References

Kaewruang, P.; Benjakul, S.; Prodpran, T. Effect of phosphorylation on gel properties of gelatin from the skin of unicorn leatherjacket. Food Hydrocolloids 2014, 35, 694-699, doi:10.1016/j.foodhyd.2013.08.017.

  1. Line 384-387: Provide suitable reference for the statement.

Response: Thank you so much for your detailed suggestions. The modification has been made in Line 497-499 of revised manuscript.

References

Yan, S.; Xu, J.; Zhang, S.; Li, Y. Effects of flexibility and surface hydrophobicity on emulsifying properties: Ultrasound-treated soybean protein isolate. Lwt 2021, 142, doi:10.1016/j.lwt.2021.110881.

O'Sullivan, J.; Murray, B.; Flynn, C.; Norton, I. The effect of ultrasound treatment on the structural, physical and emulsifying properties of animal and vegetable proteins. Food Hydrocolloids 2016, 53, 141-154, doi:10.1016/j.foodhyd.2015.02.009.

  1. Section6: The paragraph lacks proper justification. Provide more references and precise justification for the result obtained.

Response: Thank you so much for your detailed suggestions. The modification has been made in Line 518-527 of revised manuscript.

References

Sila, A.; Martinez-Alvarez, O.; Krichen, F.; Gómez-Guillén, M.C.; Bougatef, A. Gelatin prepared from European eel (Anguilla anguilla) skin: Physicochemical, textural, viscoelastic and surface properties. Colloids and Surfaces A: Physicochemical and Engineering Aspects 2017, 529, 643-650, doi:10.1016/j.colsurfa.2017.06.032.

Jellouli, K.; Balti, R.; Bougatef, A.; Hmidet, N.; Barkia, A.; Nasri, M. Chemical composition and characteristics of skin gelatin from grey triggerfish (Balistes capriscus). LWT - Food Science and Technology 2011, 44, 1965-1970, doi:10.1016/j.lwt.2011.05.005.

  1. Line 418-419: Be very careful while writing the sentences for grammar and formatting issues. Rewrite it.

Response: Thank you so much for your detailed suggestions. The modification has been made in Line 537-538 of revised manuscript.

  1. Section 4: Be careful of grammar issues. Rewrite the conclusion.

Response: Thanks for the reviewer's good evaluation and detailed suggestions. As the reviewer’s advice, we revised the conclusion of the revised draft in Line 546-561.

Overall comments

  1. Keep the values after decimal places uniform throughout the paper.

Response: Thank you so much for your detailed suggestions. In revised manuscript, we kept two decimal places in the whole paper.

  1. Strictly follow journal’s guidelines while writing “Figure or Table” within the results and discussion section, moreover in the entire paper.

Response: Thank you so much for your detailed suggestions. In revised manuscript, the "Figure and Table" have been written according to the journal’s guidelines.

  1. Maintain a uniform format for all the references as per journal’s guidelines.

Response: Thank you so much for your detailed suggestions. In revised manuscript, all the references are kept in a uniform format according to the guidelines of journals.

To sum up, we have rechecked the full text and try our best to revise it on the basis of comments. We hope this revised manuscript has addressed your concerns. Please do not hesitate to contact us at the address below if there are any questions.

Tel: +86-15228394850

Email: [email protected].

Thank you again for your hard work!

Best wishes to you!

 Yours sincerely,

Jinmeng He

 2022-11-15

Round 2

Reviewer 1 Report

The authors have addressed majority of the questions raised by the reviewers, and only minor edits are required prior acceptance.

Please add the explanation of the 6.67% concentration of gel (add reference of similar).

The sentences started with “Add” have to be changed (Lines 183,198).

Line 219: 7 days.

Line 226-229: Improve English.

Explain GB6789-2013, the sentence is not clear.

Line 266-269: English has to be improved in sentence

Line 301-306: English has to be improved

Conclusion has to be improved in terms of English

Author Response

Cover Letter

Dear Editor and Reviewers,

With this letter we are submitting the revised version of our manuscript entitled ‘‘The structural and functional differences between three species of fish scale gelatin and pigskin gelatin’’ by Jinmeng He et al (Manuscript ID: foods-2023162). Your kind advices and those comments were all valuable and very helpful for revising and improving our paper, as well as the important guiding significance to our researches. Revised portion are marked in blue in the paper. The responses to editors and reviewers’ comments/suggestions of our manuscript have been answered carefully point-by-point as follows. Some minor mistakes within the manuscript have also been corrected. We appreciate for Editors/Reviewers’ warm work earnestly, and hope that the correction will meet with approval.

Responses to the comments of Reviewer 1:

  1. Please add the explanation of the 6.67% concentration of gel (add reference of similar).

Response: Thanks for the reviewer's detailed suggestions. We have made modifications in Line 180-181 of revised manuscript.

  1. The sentences started with “Add” have to be changed (Lines 183,198).

Response: Thanks for the reviewer's detailed suggestions. We changed the sentences started with “Add” in Line 139-140&181-182&197-199 of revised manuscript.

  1. Line 219: 7 days.

Response: Thanks for the reviewer's detailed suggestions. We have made modifications in Line 218 of revised manuscript.

  1. Line 226-229: Improve English.

Response: Thanks for the reviewer's precious suggestions. We rewrote this passage in Line 225-229 of revised manuscript.

  1. Explain GB6789-2013, the sentence is not clear.

Response: We’re sorry to make you confused. GB6783-201 is one of the Chinese National Standard also known as food additive: gelatin. We have expressed this sentence more clearly in Line 256 of revised manuscript.

  1. Line 266-269: English has to be improved in sentence

Response: Thanks for the reviewer's precious suggestions. We rewrote this passage in Line 265-268 of revised manuscript.

  1. Line 301-306: English has to be improved

Response: Thanks for the reviewer's precious suggestions. We rewrote this passage in Line 300-308 of revised manuscript.

  1. Conclusion has to be improved in terms of English

Response: Thanks for the reviewer's precious suggestions. We have revised the conclusion in Line 546-555 of revised manuscript.

To sum up, we have rechecked the full text and try our best to revise it on the basis of comments. We hope this revised manuscript has addressed your concerns. Please do not hesitate to contact us at the address below if there are any questions.

Tel: +86-15228394850

Email: [email protected].

Thank you again for your hard work!

Best wishes to you!

 Yours sincerely,

Jinmeng He

2022-11-21

Reviewer 2 Report

The Ms has been significantly improved, but there are still some problems, as detailed below.

1)         The revised abstract still does not meet the requirements. The abstract should be an overview of the article, not just a presentation of the data. Specifically, structural differences are not described in the abstract. Besides, functional differences should also be summarized to provide guidance.

2)         Line 68, It is obvious that among different substances have the structural and functional differences. So this Ms should focus on the relationship between structure and function, or how does structure determine function.

3)         Authors still need to take some problems seriously with the format of the citation in full text, such as in Line 58-66, there is a major contradiction between the format here of the citatio and what appears in Line 85 and so on. Besides, note the unit format that appears for example in Line 142-148.

4)         Line 258, The quotation here is improper. Chinese national standards should be translated and quoted.

5)         For example, Need spaces between numbers and units in Fig.2.

Author Response

Cover Letter

Dear Editor and Reviewers,

With this letter we are submitting the revised version of our manuscript entitled ‘‘The structural and functional differences between three species of fish scale gelatin and pigskin gelatin’’ by Jinmeng He et al (Manuscript ID: foods-2023162). Your kind advices and those comments were all valuable and very helpful for revising and improving our paper, as well as the important guiding significance to our researches. Revised portion are marked in blue in the paper. The responses to editors and reviewers’ comments/suggestions of our manuscript have been answered carefully point-by-point as follows. Some minor mistakes within the manuscript have also been corrected. We appreciate for Editors/Reviewers’ warm work earnestly, and hope that the correction will meet with approval.

Responses to the comments of Reviewer 2:

  1. The revised abstract still does not meet the requirements. The abstract should be an overview of the article, not just a presentation of the data. Specifically, structural differences are not described in the abstract. Besides, functional differences should also be summarized to provide guidance.

Response: Thanks for the reviewer’s precious comments and suggestions. We have revised the abstract in Line 8-19 of revised manuscript.

  1. Line 68, It is obvious that among different substances have the structural and functional differences. So this Ms should focus on the relationship between structure and function, or how does structure determine function.

Response: Thanks for the reviewer’s precious comments and suggestions. We have made modifications in Line 66-69 of revised manuscript.

  1. Authors still need to take some problems seriously with the format of the citation in full text, such as in Line 58-66, there is a major contradiction between the format here of the citatio and what appears in Line 85 and so on. Besides, note the unit format that appears for example in Line 142-148.

Response: Thanks for the reviewer's detailed suggestions. We have made modifications in Line 57-64&142-146 of revised manuscript.

  1. Line 258, The quotation here is improper. Chinese national standards should be translated and quoted.

Response: Thank you so much for your careful check. The modification has been made in Line 256 of revised manuscript.

  1. For example, Need spaces between numbers and units in Fig.2.

Response: Thanks for the reviewer's detailed suggestions. We have made modifications in Line 337 of revised manuscript.

To sum up, we have rechecked the full text and try our best to revise it on the basis of comments. We hope this revised manuscript has addressed your concerns. Please do not hesitate to contact us at the address below if there are any questions.

Tel: +86-15228394850

Email: [email protected].

Thank you again for your hard work!

Best wishes to you!

 Yours sincerely,

Jinmeng He

2022-11-21

Reviewer 4 Report

All the major concerns have been revised.

Author Response

Dear Editor and Reviewers,

Thanks for your letter and for the reviewers' comments concerning our manuscript entitled ‘‘The structural and functional differences between three species of fish scale gelatin and pigskin gelatin’’ by Jinmeng He et al (Manuscript ID: foods-2023162). We appreciate the time and effort that you and the reviewers dedicated to providing feedback on our manuscript and are grateful for the insightful comments on and valuable improvements to our paper. We have incorporated most of the suggestions made by the reviewers, and we tried our best to improve the manuscript.

Thank you again for your hard work!

Best wishes to you!

Yours sincerely,

Jinmeng He

2022-11-21